# Gallium Oxide for Gas Sensor Applications: A Comprehensive Review

**DOI:** 10.3390/ma15207339

**Published:** 2022-10-20

**Authors:** Jun Zhu, Zhihao Xu, Sihua Ha, Dongke Li, Kexiong Zhang, Hai Zhang, Jijun Feng

**Affiliations:** 1School of Physical Science and Technology, Inner Mongolia University, Hohhot 010021, China; 2Global Zero Emission Research Center (GZR), National Institute of Advanced Industrial Science and Technology (AIST), Tsukuba 3058560, Japan; 3College of Sciences, Inner Mongolia University of Technology, Hohhot 010051, China; 4ZJU-Hangzhou Global Scientific and Technological Innovation Center, School of Materials Science and Engineering, Zhejiang University, Hangzhou 311200, China; 5School of Microelectronics, Dalian University of Technology, Dalian 116602, China; 6Shanghai Key Laboratory of Modern Optical System, Engineering Research Center of Optical Instrument and System (Ministry of Education), School of Optical-Electrical and Computer Engineering, University of Shanghai for Science and Technology, Shanghai 200093, China

**Keywords:** Ga_2_O_3_, electric properties, preparation methods, gas sensors, enhancement strategies

## Abstract

Ga_2_O_3_ has emerged as a promising ultrawide bandgap semiconductor for numerous device applications owing to its excellent material properties. In this paper, we present a comprehensive review on major advances achieved over the past thirty years in the field of Ga_2_O_3_-based gas sensors. We begin with a brief introduction of the polymorphs and basic electric properties of Ga_2_O_3_. Next, we provide an overview of the typical preparation methods for the fabrication of Ga_2_O_3_-sensing material developed so far. Then, we will concentrate our discussion on the state-of-the-art Ga_2_O_3_-based gas sensor devices and put an emphasis on seven sophisticated strategies to improve their gas-sensing performance in terms of material engineering and device optimization. Finally, we give some concluding remarks and put forward some suggestions, including (i) construction of hybrid structures with two-dimensional materials and organic polymers, (ii) combination with density functional theoretical calculations and machine learning, and (iii) development of optical sensors using the characteristic optical spectra for the future development of novel Ga_2_O_3_-based gas sensors.

## 1. Introduction

Gallium oxide (Ga_2_O_3_), as one type of ultrawide bandgap (UWBG) semiconducting material [1], has received tremendous attention ever since 2012 when Higashiwaki et al. successfully developed the first single-crystal Ga_2_O_3_ field-effect transistors (FETs) [2]. Over the past decade, Ga_2_O_3_ has found main applications in power electronics, solar-blind ultraviolet (UV) photodetectors, and radiation detectors, as well as gas sensors [3,4]. A variety of electronic and optoelectronic devices such as Schottky barrier diodes (SBDs) [5,6] and FETs, including MESFETs, MOSFETs, MODFETs, and HEMTs [6,7,8,9] based on Ga_2_O_3_ bulk single crystals, thin films, and nanostructured materials, have been achieved thanks to the advance in growth and characterization technologies and the unique properties of Ga_2_O_3_. There have been tens of review articles [5,6,7,8,9,10,11,12,13,14,15,16,17,18,19,20,21,22,23,24,25,26,27,28,29,30,31,32,33,34,35,36,37,38,39,40] concerning Ga_2_O_3_ that cover the growth techniques, the physical and chemical properties, and the state-of-the-art device fabrications. Although initial studies on the gas-sensing properties of Ga_2_O_3_ thin films were launched by Fleischer and Meixner [41,42] in the early 1990s, few review papers on Ga_2_O_3_-based gas sensors exist in the literature. Except for a complete review [26,27] that focuses on the gas sensors made by β-Ga_2_O_3_ nanowires and thin films, the related review can be only found in several works [4,14,15], which are usually not specialized in the domain of sensors.

It is known that Ga_2_O_3_ is a very important gas-sensing material widely used for monitoring exhaust gases of automobiles, flue gases of incinerators, pollutant gases of refinery plants, and explosive gases from military applications [43]. The exploration of Ga_2_O_3_-based gas sensors has never broken off in the last three decades, and more than five publications on average were present every year, as evident from Figure 1. In this article, we give a comprehensive overview on the distinctive aspects of Ga_2_O_3_ for gas sensor applications. The rest of the article is organized as follows. The polymorphs and crystal structures of Ga_2_O_3_ are presented in Section 2, followed by an introduction of the basic electrical properties of Ga_2_O_3_ in Section 3. The preparation methods used for fabricating Ga_2_O_3_-sensing material are provided in Section 4. Four key aspects in Ga_2_O_3_-based gas sensors, i.e., sensing mechanisms, evaluation criteria, classification of typical sensors, and performance enhancement strategies, are discussed in Section 5. Finally, a brief conclusion and outlook will be described in the last section.

## 2. Polymorphs and Crystal Structures of Ga_2_O_3_

In the context of crystallography, polymorphism is the occurrence of different crystal structures for the same chemical entity [44]. Roy et al. [45] first identified the polymorphs of Ga_2_O_3_ by X-ray diffraction (XRD) when studying the phase equilibriums of the Al_2_O_3_-Ga_2_O_3_-H_2_O system in 1952. They reported that there were five polymorphs of Ga_2_O_3_, labeled as α, β, γ, δ, and ε. The polymorphs were further confirmed by Yoshioka et al. [46] via first-principle calculations and by Zinkevich and Aldinger [47] via thermodynamic theoretical calculations. The Ga-O binary phase diagram has been also established for the first time in the work of Zinkevich and Aldinger [47]. Playford et al. [48] and Cora et al. [49] discovered the new κ-phase of Ga_2_O_3_ with a mixture of the β- or ε-phase. The electronic structures and polar properties of κ-Ga_2_O_3_ were demonstrated using density functional theory (DFT) [50]. Single-domain κ-Ga_2_O_3_ thin films have been successfully grown on the ε-GaFeO_3_ substrate by Nishinaka et al. in 2020 [51]. XRD *φ*-scan and transmission electron microscopy (TEM) revealed that the as-grown κ-Ga_2_O_3_ thin films comprised a single-domain, and none of the in-plane rotational domains were present in the films.

A summary of these six polymorphs observed in Ga_2_O_3_ until now is presented in Table 1. Among these polymorphs, β-phase is the most stable thermodynamically. All the other polymorphs are metastable and will transform into β-phase over a certain temperature, as shown in Figure 2. The interconversion of other Ga_2_O_3_ polymorphs possibly occurs under different temperatures and pressures. The formation-free energies of all the phases except κ follow the β < ε < α < δ < γ order at a low temperature [46]. Figure 3a depicts the schematic crystal structure of each polymorph. Almost all the phases demonstrate anisotropy. Taking β-Ga_2_O_3_ as an example, it is clearly seen from Figure 3b that the atomic arrangements for each crystal plane are different, which leads to different atomic configurations and dangling bond densities and therefore nonequivalent sensing properties along different crystal orientations [52].

## 3. Electrical Properties of Ga_2_O_3_

It is of interest to understand the electrical properties of Ga_2_O_3_ that play a critical role in determining the operation and functionality of a gas sensor.

As known, the electrical conductivity (*σ*) of a semiconductor has a relationship with the carrier concentration (*n*) and carrier mobility (*μ*), which are written as
*σ* = *enμ*,(1)
where *e* is the elementary charge.

Theoretically, undoped stoichiometric Ga_2_O_3_ is a transparent insulator because of its ultrawide bandgap of ~4.9 eV. However, the as-prepared Ga_2_O_3_ exhibits intrinsic n-type conductivity, and the reason for the unintentional doping is still under debate. Oxygen vacancies have been considered as a source of the intrinsic electrical conductivity for a long time [53,54]. Varley et al. [55] suggested that oxygen vacancies should act as deep donors that cannot directly account for the intrinsic electrical conductivity, because the ionization energy of such vacancies was calculated as more than 1 eV. It has been widely accepted that residual impurities such as Si and H with lower formation energies in the growth process have possibly given rise to the underlying conductivity of Ga_2_O_3_ since then [14,15,29]. By doping impurities such as Sn, Si, and Ge in Ga_2_O_3_, the free electron concentration could reach the magnitude of 10^19^ cm^−3^ in the bulk crystals and 10^20^ cm^−3^ in the case of thin films at room temperature [25,28,29,37]. In general, the ionization energy of the shallow donors has a value of less than 70 meV, which decreases with the dopant concentration. For instance, the ionization energy for Sn ranges from 7.4 meV to 60 meV; for Si, ranges from 16 meV to 50 meV; and for Ge, ranges from 17.5 meV to 30 meV [29]. The free electron concentration can be also increased by the growth conditions and post-thermal treatment [15]. However, p-type doping of Ga_2_O_3_ is still a huge challenge.

So far, the measured room temperature Hall mobility in β-Ga_2_O_3_ reaches 184 cm^2^/Vs [56], lower than the theoretical predicted value of 300 cm^2^/Vs [2]. The electron mobility varies as functions of the temperature and carrier density. Fleischer and Meixner [57], Irmscher et al. [58], Ma et al. [59], and Gato et al. [60] studied the temperature dependence of the electron mobility of single-crystal and polycrystalline Ga_2_O_3_ through Hall effect measurements. As shown in Figure 4, both the electron density and the Hall mobility increase with the increasing temperature. At an elevated temperature, the electron mobility in both cases is not determined by grain boundaries but the crystal lattice itself [57]. The intrinsic mobility is limited by optical phonon scattering at a high temperature and by ionized impurity scattering at a low temperature [59]. Moreover, the electron mobility falls when the electron density increases. The electronic effective mass at low and moderate free electron concentrations is estimated at about 0.28 of the free electronic mass, and the static dielectric constant is near 10 [61]. Table 2 highlights the basic electric properties of β-Ga_2_O_3_ reviewed in the literature. It should be pointed out that these material parameters, including electronic mobility, electronic effective mass, and dielectric constant, are greatly modified by the crystal anisotropy effect.

## 4. Preparation Methods of Ga_2_O_3_ Gas-Sensing Material

Owing to its high chemical and thermal stability, Ga_2_O_3_ is regarded as an important n-type gas-sensing material [43]. Amidst six polymorphs, β-Ga_2_O_3_ is the most favorable modification for constructing gas sensors. Sometimes, the amorphous [62,63,64] and metastable phases [65,66] of Ga_2_O_3_ have also been proposed for the purposes of detecting oxygen, ozone, and so on. Nonstoichiometric amorphous film with certain degree of oxygen vacancy leads to a lower electric conductivity and, thus, a faster rising response and a higher sensitivity to O_2_ [62,63]. The amorphous Ga_2_O_3_ thin films, which were decorated with InGaZnO nanoparticles, showed a boosted room temperature-sensing capability of ozone [64]. α- and ε (κ)-Ga_2_O_3_ thin films demonstrated sensing properties towards a large amount of oxidizing and reducing gases [65,66]. Usually, bulk crystals; thin films; and nanostructures (e.g., nanowires, nanorods, nanoparticles, etc.) have been prepared for various types of gas sensors. In this section, particular emphasis is placed on the commonly used growth techniques to prepare Ga_2_O_3_-sensing material.

### 4.1. Bulk Single-Crystal Growth

As for gas sensor applications, a few melt growth techniques have been used for preparing bulk β-Ga_2_O_3_ single crystals. These techniques contain the floating zone (FZ) [67,68,69], Czochralski (CZ) [70,71], and edge-defined film-fed (EFG) [52] growths. At high temperatures, β-Ga_2_O_3_ single crystal shows an oxygen-sensing property by the electrical resistance measurement. Oxygen diffusion in bulk takes place more likely by interstitial migration rather than by vacancy migration. The hydrogen-sensing characteristics of Pt Schottky diodes have been investigated using ($\bar{2}$01) and (010) β-Ga_2_O_3_ single crystals grown by the FZ and EFG methods. Different gallium and oxygen atomic configurations of Ga_2_O_3_ surfaces result in somewhat distinctive hydrogen responses of ($\bar{2}$01) and (010) facets. However, it remains unknown which is the chemical active crystal face of β-Ga_2_O_3_. The carrier mobility in single-crystal and polycrystalline ceramics was shown to be identical by the high-temperature Hall measurements [57]. Therefore, Ga_2_O_3_ polycrystalline thin films and nanostructures have become more popular than bulk crystals for advanced gas sensors. Since there have been quite a bit of research on gas-sensing characteristics of β-Ga_2_O_3_ single crystals, the bulk crystal growth methods of Ga_2_O_3_ are not discussed in detail here and can be found in other reviews [3,14,15,16,36]. Numerous approaches have been employed for the preparation of Ga_2_O_3_-sensing materials in the form of thin film and nanostructures. Later on, we will focus our discussion on the four most prevalent synthesis routes in the literature: magnetron sputtering, chemical vapor deposition (CVD), sol–gel synthesis, and hydrothermal synthesis.

### 4.2. Magnetron Sputtering

One of the most popular techniques for the growth of Ga_2_O_3_-sensing material is magnetron sputtering because of its low cost, simplicity, and low operating temperature. It is a physical vapor deposition technique to deposit thin films in which atoms are removed from a sintered target by bombardment with positive gas ions such as Ar^+^. A magnetic field is added beneath the target to create plasma at lower working pressures and to deflect and confine electrons. Since gallium has a low melting point of 29.8 °C, Ga_2_O_3_ thin films can be sputtered only from the metal oxide target. A radio frequency (RF) voltage is usually applied in order to prevent the target from charging up due to the bombardment from positively charged ions, as the Ga_2_O_3_ target is an insulating ceramic one. Saikumar et al. [24] provided a review of RF-sputtered films of Ga_2_O_3_ and clarified the influences on the crystallinity and film properties of Ga_2_O_3_ from various sputtering parameters, such as substrate temperature, sputtering power, annealing temperatures, and oxygen partial pressure in the reactive sputtering.

Different substrates such as sapphire [72,73,74,75,76,77,78,79,80,81,82,83,84,85,86,87,88], BeO [41,42,89,90,91,92,93,94,95,96,97], silicon [98,99,100,101], and quartz glass [101,102,103,104] frequently equipped with an interdigital electrode have been chosen to deposit Ga_2_O_3_-sensing layers using the sputtering technique. The obtained films are polycrystalline or amorphous Ga_2_O_3_ thin films. After post annealing, the sputtering β-Ga_2_O_3_ polycrystalline films are very suitable for making high-temperature sensors owing to their low-term chemical stability. Ga_2_O_3_ with incomplete crystallinity or the amorphous phase has also aroused interest in a room temperature ozone sensor due to the large base resistance and oxygen vacancies [64]. Figure 5 shows the AFM images and XRD patterns of the Cr-doped β-Ga_2_O_3_ thin films synthesized on the sapphire substrates by means of RF magnetron sputtering of the Ga_2_O_3_ target in oxygen–argon plasma. After annealing, the Ga_2_O_3_ thin films with and without the addition of Cr dopant had a β-phase polycrystalline structure, and the average grain size was estimated to be 90–100 nm for pure β-Ga_2_O_3_ and 25–50 nm for Cr-doped β-Ga_2_O_3_. However, the main drawback for gas sensor applications is that the sputtered films are too compact to increase the specific surface area.

### 4.3. Chemical Vapor Deposition

Chemical vapor deposition (CVD) is particularly interesting not only because it gives rise to high-quality Ga_2_O_3_ thin films and nanostructures but also because it is applicable to large-scale production [105]. CVD, also referred to as chemical vapor transport or vapor phase epitaxy, is a conventional method often using a multitemperature zone tubular furnace (see Figure 6a) or a stainless steel chamber as a reactor. As metal organic precursors are used, the CVD technique is called MOCVD, while, in the case of hydride or halide precursors, the technique is named HVPE. Strictly speaking, thermal oxidation should not belong to the CVD technique. It will be discussed together in this section due to the high growth temperature and chemical reaction in an activated gaseous environment similar to CVD. In the CVD process, the reactant gases are diluted in carrier gases and introduced into the reaction chamber or region. When the reactant gases approach the surface of the heated substrate, chemical reactions occur on the surface to form the deposited material. The energy necessary to start the desired chemical reaction can be supplied as thermal energy or photon energy or glow discharge plasma. Gaseous reaction by-products are then evacuated with the carrier gas by a rotary pump.

For the growth of Ga_2_O_3_-sensing material, liquid gallium [106,107,108,109,110,111,112,113,114,115,116,117,118,119,120,121,122,123], gallium nitride [124,125,126,127,128,129,130,131,132], gallium sulfides, including Ga_2_S_3_ [133] and GaS [134], gallium acetylacetonate [135], and gallium chloride [65,66,136,137,138] are often used as the gallium source, while pure oxygen or water vapor is selected as the oxygen source. Argon or nitrogen is used as the carrier gas. Sometimes, a mixture of Ga_2_O_3_ and carbon powders was also used as the precursor material to fabricate Ga_2_O_3_ nanostructures [139,140,141]. The key growth parameters, including precursors, growth temperature, growth pressure, growth duration, and gas flow rate, as well as separation distance between the metal source and substrate, all play important roles in depositing Ga_2_O_3_. Mostly, β-Ga_2_O_3_ nanostructures in the forms of nanowires, nanobelts, and nanorods will be possibly obtained, determined by the vapor–liquid–solid (VLS) or vapor–solid (VS) growth mechanisms [11]. Figure 6b–e shows some newly CVD-deposited Ga_2_O_3_ nanostructures, such as nanowires, nanorods, nanobelts, nanoribbons, and core–shell nanowires in recent years. Although a large variety of Ga_2_O_3_ nanostructures can be grown epitaxially on substrates using CVD, its disadvantage is that a high operating temperature is required, and the by-product gas may be hazardous and harmful.

**Figure 6 materials-15-07339-f006:**
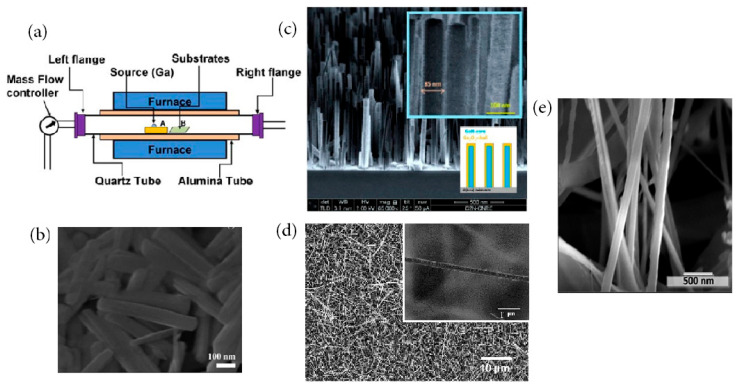
(**a**) Schematic experimental setup of the CVD furnace for the growth of Ga_2_O_3_ nanostructures [11]. Copyright 2013 Wiley. (**b**) SEM image of β-Ga_2_O_3_ nanorods [135]. Copyright 2016 The Royal Society of Chemistry. (**c**) SEM image of GaN/Ga_2_O_3_ core–shell nanowires [131]. Copyright 2019 MDPI. (**d**) SEM and HR-TEM images of β-Ga_2_O_3_ nanowires [140]. Copyright 2020 Elsevier. (**e**) SEM image of SnO_2_-coated β-Ga_2_O_3_ nanobelts [132]. Copyright 2021 Elsevier.

### 4.4. Sol–Gel Synthesis

Of all the solution synthesis approaches, the sol–gel method is the most extensively used method to synthesize metal oxide-sensing material, since it allows for exquisite control over the size, shape, and crystal phase of the resultant material. The sol–gel process usually involves several steps: (i) dispersion of colloidal particles in a liquid to form a sol, (ii) deposition of the sol solution on a substrate by spraying or dipping or spinning, (iii) polymerization of the particles in the sol to become a gel by stabilizing the removal of the component, and (iv) pyrolysis of the remaining organic or inorganic components, thus forming the final film. Inorganic metal salts or metal organic compounds, e.g., gallium nitrate [142,143,144,145] and gallium isopropoxide [146,147,148,149,150,151,152], were commonly available precursor solutions used in the sol–gel process of Ga_2_O_3_-sensing material. The sol–gel synthesis of Ga_2_O_3_ films depends on the solvent, pH value, viscosity, temperature, and so on. The surface morphology and crystallinity can be modified by subsequent heat treatment. However, disadvantages such as weak adhesion and low wear resistance limit its application in sensor fabrication.

### 4.5. Hydrothermal Synthesis

Due to easy operation and tunable growth parameters, hydrothermal synthesis is an important approach for the preparation of nanocomposites for gas sensor application. The hydrothermal process begins with an aqueous mixture of soluble metal salt precursors. Then, the solution is placed in an autoclave for reaction under relatively high pressure and moderate temperature conditions. In most cases, gallium nitrate hydrate Ga(NO_3_)_3_•*x*H_2_O [153,154,155,156,157,158,159,160,161,162] was used as the precursor for synthetizing Ga_2_O_3_ nanomaterials. The chosen solution has been always distilled water, while other organic solvent such as alcohol [156,157] was also used. After hydrothermal growth, the precipitates were calcinated for a couple of hours to improve the crystallinity. To obtain Ga_2_O_3_-sensing material with a certain size and morphology, it needs to precisely modulate the PH value and concentration of the solution, temperature, pressure, and reaction time. However, it is difficult to control the tailored phases and exact morphologies. Pilliadugula and Krishnan [159] studied the effect of PH on the surface morphology of hydrothermal synthesized β-Ga_2_O_3_ powders. Figure 7 shows the morphology evolutions of the as-prepared samples at different PH values. It was demonstrated that an extremely alkaline solution (PH = 14) caused higher-order hierarchical structures, whereas the acidic solution (PH = 5) facilitated nanorod structures. Cocoon-shaped morphology was formed as the PH value was increased from 7 to 11.

### 4.6. Other Methods

Besides the above-discussed methods, there have been other vacuum and nonvacuum growth techniques, such as pulsed laser deposition (PLD) [163,164,165,166,167], atomic layer deposition (ALD) [168], coprecipitation [169,170], and spray pyrolysis [171], for the fabrication of Ga_2_O_3_-sensing material. Photoelectrochemical oxidation [172,173,174] was also used to grow Ga_2_O_3_ thin films on the GaN surface. In this process, the GaN epitaxial film was dipped into a phosphoric acid solution and then oxidized under the illumination of a He-Cd laser source with wavelength of 325 nm. An amorphous Ga_2_O_3_ film was directly grown and could be converted to the β-Ga_2_O_3_ phase by annealing in O_2_ or N_2_ ambiance. Recently, a novel liquid gallium-based sonication approach [175,176,177] was developed to prepare Ga_2_O_3_ thin films with lots of microparticles. Using this approach, ternary alloy oxides with tuning compositions were achieved by the probe sonication of liquid gallium with traces of In, Sn, and Zn in the water medium or other solvents. After annealing, the as-prepared samples can be applied for sensing NO_2_ at a low temperature.

## 5. Ga_2_O_3_-Based Gas Sensors

Over the past three decades, several types of gas sensors using Ga_2_O_3_ bulk crystals, thin films, and nanomaterials have been developed to detect diverse oxidizing and reducing gases during a wide operational temperature range from room temperature to more than 1000 °C. In this section, we will focus on the advances in Ga_2_O_3_-based gas sensors, which cover the fundamentals of semiconductor metal oxide gas sensors and the prevailing strategies for how to improve the performance of Ga_2_O_3_ gas sensors.

### 5.1. Sensing Mechanisms

Gas-sensing mechanisms explain why the gas can cause changes in the electrical properties of a sensor. Several common gas-sensing mechanisms of metal oxide semiconductor gas sensors were reviewed by Ji et al. [178] in detail. However, the mechanisms governing Ga_2_O_3_-based gas sensors seem quite different. Most gas sensors using Ga_2_O_3_ are based on the resistive or conductive change upon target gas exposure. The sensing mechanisms of conductivity for Ga_2_O_3_-based gas sensors can be found in Refs. [3,26,179,180,181]. As mentioned above, the conductance of n-type Ga_2_O_3_ is determined by the carrier concentration and electron mobility. Both of them are temperature-dependent. When detecting gas, the change in conductance of Ga_2_O_3_ is dominated by the variation of electron density at a high temperature, while the influence of the conductance of Ga_2_O_3_ from electron mobility at a low temperature is more distinctive. As far as the mechanism that describes the interaction between the target gas and the sensitive metal oxide is concerned, three temperature-dependent regimes can be distinguished for Ga_2_O_3_-based resistive gas sensors [4,26,179,180,181,182] from Figure 8a.

In a high operating temperature range above 800 °C, the oxygen in the surrounding atmosphere and in the crystal lattice is in dynamic equilibrium, which means the oxygen exchange undergoes constantly between the bulk lattice of Ga_2_O_3_ and the ambient. If there is a reduction in the proportion of oxygen in the surrounding atmosphere, Ga_2_O_3_ crystal will experience an increase in the concentration of positively ionized oxygen defects in the lattice. Using the Kröger–Vink notation, the sum of the processes can be described as [180]
(2)$${\mathrm{Ga}}_{2}O_{3}↔2{\mathrm{Ga}}_{\mathrm{Ga}}^{x}+2O_{O}^{x}+V_{O}^{*}+e^{-}+1/2O_{2}^{\left(g\right)}.$$

As a consequence, the conductivity of Ga_2_O_3_ increases because of the delocalized electrons in crystal lattice. It can be said that this sensing regime is dominated by bulk oxygen defects. One can use it to implement high temperature oxygen sensors, which obey a typical power law [26,99,183]
(3)$$\sigma \propto p_{O_{2}}^{m}e^{-E_{A}/kT},$$
where *p*_O2_ is the oxygen partial pressure and *E*_A_ is thermal activation energy of a dopant, *k* is the Boltzmann constant, and *T* is the temperature. The first part in above equation represents the contribution from the oxygen partial pressure, and the last part reveals the temperature dependence of the conductivity on the doped specimens.

In the intermediate operation temperature range below 800 °C, only an exchange of oxygen near the surface of Ga_2_O_3_ with the surrounding gas takes place. If a certain reducing gas such as CH_4_ approaches the surface, it will be oxidized by oxygen from the near-surface region, leading to the formation of oxygen vacancy donors at surface and a greater conductivity due to the release of conduction electrons. The process then will be recovered in the absence of reducing gas, since the loss of oxygen of the Ga_2_O_3_ surface is compensated by the oxygen from the ambient atmosphere. In this case, the sensing regime is mainly caused by surface oxygen defects.

At even lower temperatures, the oxygen defect equilibrium disappears. The gas-sensing behavior is dominated by the change in electron mobility controlled by grain boundaries, rather than the change in oxygen defect-affected charged carrier density. As shown in Figure 8b, on the one hand, similar to other metal oxides, a potential barrier is formed at the grain boundary of Ga_2_O_3_ by pre-adsorbed oxygen ions with negative charge in the air, leading to a charge depletion region between grains. The adsorbed oxygen ions are O_2_^−^ below 200 °C, O^−^ in the range of 200–550 °C, and O^2−^ above 550 °C. The width of charge depletion region is determined by the Debye length [26]:(4)$$L_{D}=\sqrt{\varepsilon kT/ne^{2}},$$
where *ε* is the permittivity, *n* is the electron density, and *e* is the electronic charge.

When exposed to a reducing gas such as CO, the gas molecules react with chemisorbed oxygen ions at the Ga_2_O_3_ surface and return the captured electrons to the conduction band, resulting in the decrease of the depletion width. Hence, the conductance is increased due to the easier electronic transport across the grain boundary. In the case of the presence of an oxidizing gas such as NO_2_, a competitive adsorption process may take place, and the overall results are a wider depletion region, smaller electron mobility, and a decrease of conductivity. However, the sensitivity toward NO_2_ could be very low, since the Debye length of Ga_2_O_3_ is as large as several μm [181].

On the other hand, reducing a gas such as H_2_ will be chemisorbed at the surface via covalent bonds to form adsorbed molecules with positive charges. To satisfy the electric neutrality, more conduction electrons are released, yielding a conductive increase in n-type Ga_2_O_3_. The chemisorption process happens even with no oxygen in the ambient. However, the overall concentration of the surface adsorbed species is subject to the Weisz limit [184]. Clearly, gas chemisorption and reaction at the surface play an important role in this low-temperature-sensing regime.

It should be noted that there are no upper and lower temperature limits for the three gas-sensing regimes. The modulation of the conductance of Ga_2_O_3_ by the target gas can be a consequence of one or a combination of grain boundaries, gas absorption and reaction, and oxygen vacancies. There have also been various Ga_2_O_3_-based nonresistive gas sensors using different operating principles. For example, the gas-induced changes in work function, in capacitance, in the Schottky potential barrier, and in the ionic conductivity can be used for sensing a wide range of target gases. These gas-sensing mechanisms are similar to those of other semiconductor metal oxide sensors, which can be easily found in a lot of the literatures such as [185].

### 5.2. Evaluation Criteria

Usually, the performance of Ga_2_O_3_-based gas sensors can be evaluated with respect to the ‘4s’-criteria: sensitivity, speed, selectivity, and stability [43].

The sensitivity S, also named as response, in the presence of gases is usually defined in several different forms. It is generally calculated from the ratio of the electrical readout (resistance *R* or conductance G and current *I* or volt *V*) of the sensor in background gas (usually air) *Y*_a_ to that upon exposure to certain concentrations of the target analyte *Y*_g_:(5)$$S=Y_{a}/Y_{g},$$
for oxidizing the target gas and
(6)$$S=Y_{g}/Y_{a},$$
for reducing the target gas.

Other expression forms of sensitivity that describes the variation degree of the output signal are also in use:(7)$$S=\Delta Y_{a}/Y_{g}.$$

The detection speed is evaluated by the response and recovery time. The response time is defined as the time required for a sensor to reach 90% of the total response upon exposure to the target gas. Recovery time is defined as the time required for a sensor to return to 90% of the original baseline signal upon removal of the target gas.

The selectivity of a sensor describes how much the sensor is disturbed by interfering gases from the target gas. The selectivity *Q*, which reflects the ability of a sensor to differentiate between the target gas x and the other components in the gaseous environment x′, can be expresses as:(8)$$Q=S_{x}/S_{x^{\prime }}.$$

Obviously, larger *Q* means better selection of target gas and stronger resistive to interfering gas.

The stability (reproducibility) describes the endurance of a gas sensor to maintain its output signal over a long period of time and/or to the analyte gas of varying concentrations. The stability is greatly affected by thermal aging and gaseous poisoning of the sensor layer, especially when operating in a harsh environment.

Additionally, the operating temperature, power consumption, size, and cost are the other concerns that should be considered, depending on particular applications of gas sensors. It is difficult to achieve all the optimal results of the above performance parameters at the same time. Therefore, to keep the balance according to the specific situation and requirements has become a major aim for the development of Ga_2_O_3_ gas sensors.

### 5.3. Classification of Ga_2_O_3_-Based Gas Sensors

According to the famous Yamazoe model [186], a gas sensor can be considered as an integration of a receptor and a transducer. The former is provided with a material or a material system that interacts with a target gas and thus induces a change in its own properties or emits heat or light. The latter is a device to transform such an effect into an electrical signal. There are various approaches used for gas sensor classification. In terms of transduction principles, the gas sensors that many groups have always developed using Ga_2_O_3_ as sensing materials can be classified into three categories: electrical gas sensors, electrochemical gas sensors or solid electrolyte-based gas sensors, and optical gas sensors [187].

#### 5.3.1. Electrical Gas Sensors

Electrical gas sensors, operating due to electronic conduction induced by a surface interaction with target gas, contain resistor-type gas sensors and nonresistive sensors such as SBD-, FET-, and capacitor-type gas sensors.

(1) Resistor-type gas sensors are the conductometric sensors that measure the change in resistance caused by the interaction between the sensing element and analyte gas. They possess advantages such as simple configuration, easy fabrication, and cost effectiveness and can be easily miniaturized and integrated on a microelectronic mechanical system platform. A typical resistor-type gas sensor based on Ga_2_O_3_ is depicted in Figure 9a. Pt and Au are commonly used as measuring electrodes, since the electron affinity of Ga_2_O_3_ is as large as about 4 eV [40]. Similar to solar-blind ultraviolet photodetectors, the interdigitated electrode geometry, which enables a wide contact area, is the most widely accepted geometry for a resistor sensor. In most cases, it forms the electrodes first and then deposits the Ga_2_O_3_-sensing layer on them, thereby causing no damage to the sensing material. Additionally, a Pt heater is placed on the back side of the substrate to heat the sensor.

In early years, Fleischer and Meixner in Siemens AG and their cooperators adopted the RF magnetron sputtering technique to fabricate β-Ga_2_O_3_ polycrystalline thin films and constructed resistor sensors to detect a variety of gases, such as O_2_ [41,42,74,79,96,102,188,189,190]; O_3_ [81]; H_2_ [74,78,79,82,89,90,95,191,192,193,194]; CO [75,79,82,89,91,97,188,189,191,192]; NO [78,79,96]; NH_3_ [79,96]; and hydrocarbons (HCs) such as CH_4_ [73,77,78,79,82,94,97,102,188,189,195], C_3_H_8_ [79,82,196], and C_4_H_8_ [79,188], as well as volatile organic compounds (VOCs) such as C_2_H_6_O [78,79,97,195] and C_3_H_6_O [78,80,82]. The operating temperature ranges from 1100 °C to 400 °C. Typically, polycrystalline Ga_2_O_3_ thin films can be used either for sensing oxygen (>900 °C) or reducing gases (<900 °C) [14,91]. The co-adsorption of H_2_O and other coexisting gases of the sputtered films were considered by Giber et al. [197], Reti et al. [198,199,200], and Pohle et al. [201]. Due to a high operating temperature, a self-cleaning effect was observed on the sensor surface, and the unwanted species could be eliminated to a large extent. Varhegyi et al. [202] investigated the influence from corrosive gases such as Cl_2_ and SO_2_ on the Ga_2_O_3_-sputtered layer and found that Ga_2_O_3_ was more resistant against SO_2_ but had a very fast reaction with Cl_2_ at 800 °C. A screen-printing technique was developed by Frank et al. [188], Pole et al. [201], Wiesner et al. [195], and Biskupski et al. [196] to prepare porous Ga_2_O_3_ thick films for detecting oxidizing gases such as CO_2_ and O_3_ and inflammable gases such as C_4_H_10_ and C_3_H_8_, as well as VOCs. Additionally, many other groups such as Macri et al. [72], Ogita et al. [62,63,97,103], Hovhannisyan et al. [203], Dyndal et al. [204], Almaev et al. [86,87], Manandhar et al. [101], and Sui et al. [64] studied the gas sensitivities of resistor sensors with sputtering films for O_2_, H_2_, CO, C_7_H_8_, C_2_H_6_O, and C_3_H_6_O. However, it is very hard to reduce the operating temperature of these resistor sensors using the Ga_2_O_3_ compact films to a lower value.

With the progress of advanced synthesis technologies, many novel Ga_2_O_3_ nanomaterials with different compositions and microstructures have been prepared for resistor sensors. The morphology of these Ga_2_O_3_ functional nanomaterials varies from one dimensional to three-dimensional nanostructures such as nanospheres, nanoflowers, nanowires, nanorods, nanobelts, and so on. These sensing materials showed high resistive response towards O_2_ [109], CO [109,124,129,135,140,154,155], CO_2_ [158], H_2_ [113,132,176], NH_3_ [159,162,177], NO_2_ [125,126,128,141,160,161,176], VOCs such as C_2_H_6_O [117,123,141,168], C_3_H_6_O [123,133,157], C_3_H_8_O [120], and water vapor [127,130,134], as reported by many authors. The operating temperature can be low to room temperature due to the large specific surface area and robust properties of nanostructures.

(2) Since Lundstrom et al. [205] first achieved a H_2_-sensitive Pd-gate gasFET device, there has been of great interest in semiconductor metal oxide nonresistive-type gas sensors due to the simple electrical circuit required to operate them. In contrast to the above-discussed resistor-type gas sensors, the gas-sensing properties of nonresistive ones will be less affected by the morphology of Ga_2_O_3_-sensing material. In generally, nonresistive type gas sensors consist of two types of structures, i.e., metal/semiconductor (MS) is metal/insulator/semiconductor (MIS) structures. Comparatively, the MIS structure is always adopted in designing Ga_2_O_3_ sensors with Pd or Pt as catalytic metals and Ga_2_O_3_ as reactive insulator material. There are usually three types of nonresistive-type gas sensors based on various operating principles, namely, SBD-type, FET-type, and capacitor-type gas sensors. Figure 9b–d depict the schematic device structures of these three typical Ga_2_O_3_-based nonresistive-type gas sensors. The basis with respect to SBD- or FET-type (including MIS capacitor) gas sensors with a catalytic metal is modulation in the Schottky barrier height or the flat band potential by target gas [187]. Catalytic decomposition of hydrogen on the surface of noble metal and subsequent diffusion of hydrogen atoms to the interface between metal and insulator make these devices respond to H_2_ and many hydrogen containing gases.

SBD-type sensors operate on the change in Schottky barrier height affected either by formation of a dipole layer or by modification of work function once gaseous species interact with the metal surface. Then, the gas-induced rectifying property can be recognized by I–V characteristics under a forward, as well as reverse, bias voltage. Pt/Ga_2_O_3_/SiC [146,149,150,151,152] and Pt/Ga_2_O_3_/GaN [172,173,174] Schottky diode gas sensors were built to detect H_2_ and C_3_H_6_ at different operating temperatures. Notably, Jang et al. [52] and Nakagomi et al. [68,69] reported Schottky diode gas sensors based on β-Ga_2_O_3_ single crystals, which showed an enhanced response to H_2_ and stable operation at elevated temperatures. Almaev et al. [136] studied the gas-sensing properties of Pt/α-Ga_2_O_3_: Sn/Pt Schottky metal–semiconductor–metal structures when exposed to H_2_, O_2_, CO, NO, CH_4_, and NH_3_ in the temperature range of 25–500 °C.

FET-type sensors are based on the readout of work function change of the sensing material. The gas response of FET-type sensors is measured as a shift in either gate-source voltage or drain-source current, and it has been shown that this response is related to a shift in the threshold voltage. This kind of gas sensors can detect many reducing gases, such as H_2_ and VOCs. A hybrid FET-type sensor was designed using Ga_2_O_3_ as the sensitive layer and the measured variation of the work function indicated a multiple response to H_2_, NH_3_, and NO_2_ [206,207,208]. In their work, an analytic theoretical model was proposed to explain the inner sensing mechanism. Lampe et al. [84] made a gas FET sensor in which a sputtered Ga_2_O_3_ thin film activated with Pd was used to detect CO. Stegmeier et al. [85,86] used sputtered Ga_2_O_3_ film to construct a FET-type sensor for detecting VOCs. Nakagomi et al. [118,119] reported a field–effect hydrogen sensor using Ga_2_O_3_ thin film prepared by CVD. Shin et al. [209] investigated the channel length scaling effects on the signal-to-noise the ratio of a FET-type NO_2_ sensor. It was demonstrated that the FET gas sensors had an advantage of working at room temperature.

MIS capacitor-type sensors are changes that can be made to the relative permittivity of the dielectric, the area of the electrode, or the distance between the two electrodes and, therefore, by measuring the change in the capacitance. Arnold et al. [110] reported an interdigital comb-finger structural capacitor gas sensor with β-Ga_2_O_3_ nanowires as dielectric. By capacitance measurement using a balanced AC bridge circuit, the sensor showed a rapid and reversible response to VOCs such as C_2_H_6_O and C_3_H_6_O and a more limited response to some HCs, including C_7_H_8_ at room temperature. Mazeina et al. [114,115] compared pure and functionalized Ga_2_O_3_ nanowires as active material in room temperature capacitance-based gas sensors. It was found that the functionalization of Ga_2_O_3_ nanowires with acetic acids showed a significant decrease in response to C_3_H_8_O and CH₃NO₂ as well as C₆H_15_N. In the case of pyruvic acid-functionalized nanowires, no response was observed to CH₃NO₂, but one order of magnitude increased response to C₆H_15_N was obtained compared to the pure nanowires.

#### 5.3.2. Electrochemical Gas Sensors

An electrochemical gas sensor is a device that yields an output as a result of an electrical charge exchange process at the interface between ionic or electronic conductors. Solid electrolytes exhibit high ionic conductivity resulting from the migration of ions through the point defect sites in their lattices. In solid electrolyte-based sensors, electronic conduction only makes typically less than 1% contribution to the total conductivity, while the ionic conductivity contributes to 99% of the rest. Schematic device structures for two kinds of electrochemical gas sensors containing Ga_2_O_3_ are shown in Figure 10. In an early study, NH_4_^+^ ion conducting gallate solid electrolyte was prepared by mixing K_2_CO_3_, RbCO_3_, and Ga_2_O_3_ [210]. Fabricated by the combination of NH_4_^+^-Ga_2_O_3_ and rare earth ammonium sulfate as a solid electrolyte and a solid reference electrode, the gas sensor showed outstanding NH_3_ detection in good accordance with the Nernst relation. In a mixed potential gas sensor based on O^2−^ ion conducting yttria-stabilized zirconia (YSZ) solid electrolyte, Ga_2_O_3_ was used to stabilize the metal (Au, Pt) electrode in its morphology and to inhibit its catalytic activity [211,212,213,214,215]. The results indicated that these gas sensors had a high sensitivity and good selectivity of HCs, such as C_3_H_6_. An electrochemical Pt/YSZ/Au-doped Ga_2_O_3_ impedance metric sensor was fabricated by Wu et al. [216]. The impedance of the sensor originated from the Ohmic contact resistance, the electrolyte impedance, and the interfacial impedance between the electrolyte and the sensing electrode. Strong dependence of the interfacial impedance upon the CO concentration at 550 °C was found due to an electrochemical oxidation of CO. Yan et al. [217] synthesized a mesoporous β-Ga_2_O_3_ nanoplate to make an amperometric electrochemical sensors. It was summarized that more oxygen defects and action sites in β-Ga_2_O_3_ nanoplate account for enhanced gas sensitivity in detecting CO.

#### 5.3.3. Optical Gas Sensors

Optical gas sensors detect changes in visible light or other electromagnetic waves during interactions with gaseous molecules. Reiprich et al. [122] used a corona discharge assisted growth morphology to prepare Sn-doped Ga_2_O_3_ for optical gas-sensing application. The Sn-doped Ga_2_O_3_ layer was shown to be capable of detecting small amounts of C_2_H_6_O, C_3_H_6_O, and C_3_H_8_O at room temperature. The reason was that the generation and quantity variance of negatively charged oxygen ions indirectly produced a change in the photoluminescence spectrum. It was also observed that the response toward C_3_H_6_O of a Ga_2_O_3_ layer-like structure would be increased by 30% relative to the nanowires. However, the study of Ga_2_O_3_-based optical gas sensors is in its infancy, further careful investigations are needed into this type of gas sensor.

To sum up, Ga_2_O_3_-based gas sensors exhibit broad-range sensitivity to a wide variety of analyte gases. Table 3 lists the available target gases of different types of Ga_2_O_3_-based gas sensors. The most preferable target gases are O_2_, CO, H_2_, and CH_4_ [218].

### 5.4. Performance Enhancement Strategies

As is discussed in the former section, Ga_2_O_3_ can be used to make several types of gas sensors that respond to lots of gases, ensuring a wide application range. However, Ga_2_O_3_-based gas sensors still encounter some issues such as low selectivity, average sensitivity, and high operation temperature [26,182,187,218]. Multiple ways have been developed till now by many researchers to improve the gas-sensing performance of Ga_2_O_3_-based gas sensors through material engineering and device optimization.

#### 5.4.1. Surface Modulation of Pure Ga_2_O_3_

It is well-known that the performance of Ga_2_O_3_ gas sensors greatly depends on the interaction between target gas and the Ga_2_O_3_ surface. Therefore, surface modulation, such as crystallinity, porosity, and oxygen vacancies, in the process of fabricating Ga_2_O_3_-sensing material plays an important role in determining the gas-sensing behavior. Ogita et al. [63,67,99,100,219] presented a series of studies on the influence of the annealing conditions on the high-temperature oxygen-sensing properties of the Ga_2_O_3_ thin film sensor. The results indicated that the annealing conditions affect not only the grain size, number of oxygen vacancies and surface roughness but also the oxygen-sensing properties of Ga_2_O_3_ thin films. The oxygen sensitivity of the sensors increases when the oxygen flow rate was increasing from 0 to 100% at 1000 °C. The response time of the Ga_2_O_3_ sensors decreased as the grain size increases with increasing the annealing temperature and annealing time. The authors also measured the oxygen-sensing characteristics at 1000 °C for β-Ga_2_O_3_ sputtered thin films and β-Ga_2_O_3_ single crystals and found that both sensors had response times in the same range. Grain boundaries played an important role in determining the value of the response time in sputtered films, while the surface processes primarily influenced the modifications of the resistance for single crystals. The influence of film thickness on sensing response at room temperature was investigated by Pandeeswari and Jeyaprakash [171]. The sensitivities towards 0.5 ppm and 50 ppm of NH_3_ increased as the film thickness became larger.

Due to large surface to volume ratio and abundant surface active sites to the target gas, porous and nanostructured Ga_2_O_3_ was always used as the sensing layer. Cuong et al. [113] addressed the effect of growth temperature on the microstructural properties and H_2_ sensing ability of Ga_2_O_3_ nanowires. Girija et al. [135] performed the gas-sensing analysis of morphology-dependent β-Ga_2_O_3_ nanostructure thin films. It can be seen from Figure 11 that nanorods of β-Ga_2_O_3_ had a higher surface area with a larger pore volume distribution than rectangular β-Ga_2_O_3_ structures. As a result, rod shaped β-Ga_2_O_3_ nanostructures showed better sensitivity, shorter response, and recovery time upon exposure to CO gas at 100 °C in comparison with rectangular shape β-Ga_2_O_3_ nanostructures. The improvement on sensor performance could be contributed from the high surface area, particle size, shape, numerous surface active sites and the oxygen vacancies. The effect of pH-dependent morphology evolutions on room temperature NH_3_-sensing performances of β-Ga_2_O_3_ nanostructured films was investigated by Pilliadugula and Krishnan [159]. It could be seen that the sample with hierarchical morphology and relatively low crystallite size showed utmost sensing responses at all concentrations of NH_3_ relative to the remaining samples.

High-energy crystal facet has a higher surface energy and more defect sites, thus exhibiting more active physicochemical properties and better gas-sensing performance. Jang et al. [52] demonstrated that the H_2_ responses of the (010) facet were slightly higher than that of the ($\bar{2}$01) facet when studying the sensing characteristics of Schottky diode-type gas sensors using Ga_2_O_3_ single crystal. However, it is not clear which facet is the most active one for gas sensors so far.

#### 5.4.2. Sensitizing by Noble Metal

In most of gas-sensing process, small noble metal nanoparticles or clusters distributed on the surface of Ga_2_O_3_ can improve the efficiency of catalytic reactions between target gases and the surface of sensing layers, thus remarkably enhancing the gas-sensing characteristics. At an early stage, Fleischer et al. [91], Bausewein et al. [74], and Schwebel et al. [75] studied the effects of Pt, Pd, and Au catalyst dispersions on the sensitivity to reducing gases of high temperature Ga_2_O_3_ thin film sensors. They found that Pt dispersion accelerated the response of CO, whereas significantly increased the sensitivity to H_2_. Au clusters on the Ga_2_O_3_ surface could yield a high sensitivity to CO and a distinct reduction of the cross-sensitivity towards VOCs. However, although Pt accelerated the desorption of H_2_ on Ga_2_O_3_, it had no impact on the sensitivity to H_2_. Recently, Krawczyk et al. [138] reported the impregnation of Au nanoparticles of the HVPE-prepared n-type β-Ga_2_O_3_ layer would cause an inversion to p-type conductivity at the surface. The conductive sensitivity of the Au-modified β-Ga_2_O_3_ layer toward 16 ppm of dimethyl sulfide obviously was enhanced if compared to that of the unmodified β-Ga_2_O_3_ layer. Pt and Au-decorated Ga_2_O_3_ nanowires were synthesized by Kim et al. [124] and Weng et al. [140]. Figure 12 shows the SEM and TEM images and electrical response of the gas sensors fabricated by Pt- and Au-decorated Ga_2_O_3_ nanowires. As measuring at 100 °C, the responses of the Pt-functionalized nanowire were reported to improve 27.8-, 26.1-, 22.0-, and 16.9-fold at CO concentrations of 10, 25, 50, and 100 ppm, respectively, relative to the bare Ga_2_O_3_ nanowires. However, both the response and recovery times of this Pt-functionalized CO sensor were increased significantly. Through CO gas sensor measurements at room temperature, Au-decorated single β-Ga_2_O_3_ nanowire showed better results than single pure β-Ga_2_O_3_ nanowire and multiple networked Au-decorated β-Ga_2_O_3_ nanowires. The enhancement of the CO-sensing properties was resulted from a combination of the spillover effect and the enhanced chemisorption and dissociation of the target gas.

So far, noble metals had also been used in FET-type and solid electrolyte type gas sensors to catalytically activate the sensing layer. Zosel et al. [212,213] achieved a high sensitivity for hydrocarbons in potentiometric zirconia-based gas sensors, attributed to the catalytic activity of the Au composites. Shuk et al. [215] reported that the mixed potential solid electrolyte sensor based on Au/Ga_2_O_3_ composite electrodes had a sensitivity limit of 5 ppm CO and showed good selectivity, reproducibility, and stability in the presence of other combustion gas species. The FET-type sensors thermally activated by Pd and Pt for the detection of reducing gases were studied by Lampe et al. [84] and Stegmeier et al. [85,86]. The results indicated that catalytic Pt dispersions on the micro- and nanoscale caused a remarkable gas response at room temperature to a large variety of hydrocarbons and VOCs with small concentrations (ppb-ppm). Overall, noble metals such as Pt, Pd, and Au play the roles of chemical sensitizers and electronic sensitizers in Ga_2_O_3_ sensors.

#### 5.4.3. Doping Specific Element

Doping is an effective way to influence not only the electronic properties but also the structural properties of grains, such as size and shape, leading to the enhancement of sensing behavior. Frank et al. [102,188,189] found that the doping of sputter-deposited polycrystalline Ga_2_O_3_ thin films using donor type ions such as Zr^4+^, Ti^4+^, and especially Sn^4+^ caused a modulation of the base conductivity by two orders of magnitude and strong impact on the sensitivity to reducing gases. However, there was no influence on the sensitivity to reducing gases such as CH_4_ and CO after doping Mg^2+^ in the Ga_2_O_3_ and no influence of whether donor-type or acceptor-type doping concentration on O_2_ sensitivity at high temperature. However, in the work of Almaev et al. [87,88], the authors stated that addition of Cr stimulated dissociative adsorption of O_2_ due to its high catalytic activity via a spillover mechanism, leading to a significant increase in the response of to O_2_ over the temperature range 250–400 °C. The O_2_ gas-sensing performance of Ga_2_O_3_ semiconducting thin films prepared by the sol–gel process and doped with Ce, Sb, W, and Zn was investigated by Li et al. [148]. The authors observed that the operating temperature of sensors doped with Zn reduced below 450 °C and Sensors doped with Ce had a very fast response time of typically 40 s. W-doped sensors showed the highest response to O_2_ while Sb-doped films possessed the highest base resistance. Sn was proven to be the most effective dopant in Ga_2_O_3_ for gas detection at different operating temperatures [121,122,162,170,189]. As examples, Figure 13 exhibits the enhancement of the sensor sensitivity of Sn-doped Ga_2_O_3_ gas sensors when detecting different gases. An increase in conductivity of up to two orders of magnitude, as well as an enhancement of the gas sensitivity toward CO and CH_4_, was found by employing SnO_2_ as a doping material into sputtered polycrystalline Ga_2_O_3_ thin films [189]. Doping Sn could enhance the adsorption of NH_3_ due to the higher Lewis acidity of Sn^4+^ cations than Ga^3+^ ones [170]. Moreover, the introduction of Sn causes a decrease in the average crystallite size and an increase in the specific surface area [162]. Therefore, Sn-doped Ga_2_O_3_ NH_3_ sensor showed better performance than the pure Ga_2_O_3_ one regardless of the operating temperature. Manandhar et al. [101] demonstrated that the Ti-doped nanocrystalline β-Ga_2_O_3_ films significantly accelerated the oxygen response about 20 times while retaining the stability and repeatability.

It is noted that doping can not only enhance the sensing properties of resistor-type Ga_2_O_3_ sensors but also have a great impact on design of novel sensors. Reiprich et al. [122] used Sn-doped Ga_2_O_3_ nanostructure for optical gas-sensing at room temperature. The change in the photoluminescence spectra indicated that Sn-doped Ga_2_O_3_ was capable of detecting trace amounts of VOCs such as C_2_H_6_O, C_3_H_6_O, and C_3_H_8_O. Saidi et al. [176] developed a novel liquid metal-based ultrasonication process within which additional metallic elements (In, Sn, and Zn) were incorporated into liquid Ga and then sonicated in dimethyl sulfoxide (DMSO) and water. These new types of doped Ga_2_O_3_ sensors showed very tuning sensitivities to NO_2_ and H_2_. Impedance measurements were performed on the Pt/YSZ/Au-doped Ga_2_O_3_ CO electrochemical sensor [216] and on the humidity sensor-based on Ga_2_O_3_ nanorods doped with Na^+^ and K^+^ [220]. Both types of Ga_2_O_3_ gas sensors showed fast response of target gases. Due to the synergistic effect of Ga_2_O_3_ nanorods and the doping of alkali metal ions, Ga_2_O_3_ nanorod gas sensor exhibited good linearity response and high stability over 25 days. So far, the dopants added in Ga_2_O_3_ were Zr, Ti, Mg, Sn, Ni, N, Na, K, Cr, In, Zn, Pt, and Au for the purpose of enhancing the sensing properties.

#### 5.4.4. Constructing Ga_2_O_3_ Heterostructure

Semiconducting heterostructures show great potential in gas sensors because of a high surface-to-volume and synergistic effect [221]. Once heterointerface forms, Fermi level-mediated charge transfer and band bending occur, usually resulting in a higher sensitivity. According to the types of participating semiconductors, Ga_2_O_3_ heterostructures can be divided into the n–n heterojunction and p–n heterojunction. Figure 14 illustrates the schematic diagram of the energy band structures for such kinds of heterojunctions. Many heterostructures, including heterojunctions and hierarchical heterostructures, have been developed by many researchers to improve the performance of Ga_2_O_3_ sensors. In the early years, Fleischer et al. [79,81,96,190] discovered a dramatic influence on the sensitivity and selectivity of Ga_2_O_3_ thin film sensors by sputtering different metal oxide, i.e., WO_3_, Ta_2_O_5_, NiO, AlVO_4_, CeO_2_, Sm_2_O_3_, RhO, RuO, Ir_2_O_3_, and In_2_O_3_, as the surface modification layer. These types of thin film sensors could be employed as selective oxygen sensors for gas atmospheres with an overall excess of oxygen.

As for resistor-type of gas sensors, TiO_2_, SnO_2_, ZnO, WO_3_, GaN, and GaS were used for constructing n–n heterostructures with Ga_2_O_3_ while NiO, CuO, LSFO, and LSCO of perovskite crystal structure for p–n heterostructures. Mohammadi and Fray [143] reported that mesoporous Ga_2_O_3_/TiO_2_ thin film gas sensors had response values of 13.7 and 4.3 to 400 ppm CO and 10 ppm NO_2_ gas at 200 °C, approximately 55% larger than pure TiO_2_ sensors. Jang et al. [117], Liu et al. [139], and Abdullah et al. [132] studied the sensing properties of Ga_2_O_3_/SnO_2_ core–shell nanowires and nanobelts. Compared with pure Ga_2_O_3_ nanowire sensor, the optimum sensing temperature was reduced by 200 °C for Ga_2_O_3_/SnO_2_ core–shell nanowire sensors that showed the highest ethanol response at 400 °C. Due to the fast physisorption of water and the fast formation of depletion layers caused by the large surface area of the amorphous SnO_2_ shell and Ga_2_O_3_/SnO_2_ heterojunction, Ga_2_O_3_/SnO_2_ core–shell nanoribbons had a very high sensitivity to humidity with quick response and recovery near room temperature. At 25 °C, the conductivity of this nanoribbon humidity sensor at 75% relative humidity was three orders of magnitude larger than that at 5% relative humidity. The response time and recovery time were approximately 28 s and 7 s, respectively, when the relative humidity was switched between 5 and 75%. The H_2_ gas sensor based on Ga_2_O_3_/SnO_2_ core–shell nanobelts exhibited significant enhanced performance at room temperature in terms of response, response/recovery time and repeatability. During the exposure of 100 ppm NO_2_, multiple networked Ga_2_O_3_/ZnO core–shell nanorod sensors showed a super response of 32.778% at 300 °C, which was 692 and 1791 times larger than that of bare Ga_2_O_3_ and bare ZnO nanorod sensors [126]. The sensor made by a wafer-scale ultra-thin Ga_2_O_3_/WO_3_ heterostructure [168] exhibited about 4- and 10-fold improvement in the response to C_2_H_6_O compared to that of pure WO_3_ and Ga_2_O_3_ nanofilm sensors at 275 °C. Furthermore, the Ga_2_O_3_/WO_3_ heterostructural sensor possessed a shorter response/recover time and excellent selectivity. Park et al. [129] developed a Ga_2_O_3_/GaN core–shell nanowire sensor by surface-nitridated Ga_2_O_3_ nanowire. It showed responses of 160–363% to CO concentrations of 10–200 ppm at 150 °C, which were 1.6–3.1-fold greater than those of pristine Ga_2_O_3_ nanowire sensors.

For n–n junction, an accumulation layer is usually created, whereas a depletion layer forms in p–n junction. Lin et al. [154] and Zhang et al. [161] fabricated p–n heterostructural gas sensors using perovskite-sensitized Ga_2_O_3_ nanorod arrays for CO and NO_2_ detection at high temperature. Figure 15 gives the TEM images and measured sensing performances of gas sensors made by Ga_2_O_3_/La_0.8_Sr_0.2_FeO_3_ (LSFO) and Ga_2_O_3_/La_0.8_Sr_0.2_CoO_3_ (LSCO) nanorods. Compared with the pristine Ga_2_O_3_ nanorod array sensor, close to 10 times enhanced sensitivity to 100 ppm CO was discovered for Ga_2_O_3_/LSFO sensors at 500 °C. In addition to the excellent CO sensitivity, the sensor based on Ga_2_O_3_/LSFO p–n heterostructure has a faster response time than that of the pristine Ga_2_O_3_ sensor. Ga_2_O_3_/LSCO sensors also showed nearly an order of magnitude enhanced sensitivity to 200 ppm NO_2_ at 800 °C, along with much shorter response time. Ga_2_O_3_/CuO thin films were magnetron sputtered by Dyndal et al. [204] as a gas-sensitive material for C_3_H_6_O detection measured at 300 °C. Benefited from p–n heterostructure, Ga_2_O_3_/CuO thin film sensor exhibited 40% faster response time in comparison with pure CuO one. Sprincean et al. [134] made a Ga_2_O_3_/GaS:Zn nanostructured room temperature humidity sensor, which demonstrated acceptable sensitivity on the air relative humidity in the range from 42 to 92% and stable static characteristics over 6 months.

Additionally, Wang et al. [160] investigated the gas-sensing behavior of Ga_2_O_3_/Al_2_O_3_ nanocomposite and found that the composite-based sensor had a 6.5 times higher response to 100 ppm NO*_x_* than that of the pure Ga_2_O_3_ sensor at room temperature. Notably, Sivasankaran and Balaji [177] synthesized mesoporous Ga_2_O_3_/reduced graphene oxide (rGO) nanocomposites by hydrothermal method. The results indicated that the sensing response to 200 ppm NH_3_ of Ga_2_O_3_/rGO nanocomposite was 3.7-fold larger than that of pure Ga_2_O_3_ at room temperature.

As known, Ga_2_O_3_-based Shottky diode H_2_ sensors generally has the MIS-type heterostructure in which β-Ga_2_O_3_ is served as reactive insulator and SiC [146], GaN [172,174], and AlGaN [173] are chosen as semiconductors. Lee et al. [174] concluded that the MIS-type sensor diodes exhibited better forward response than the MS-type Schottky sensor diode, because the β-Ga_2_O_3_ insulator surface provided more trap sites for hydrogen atoms at the metal–insulator interface.

#### 5.4.5. Ga-Contained Metal Oxide

Aside from doping in Ga_2_O_3_, the gas-sensing properties can be optimized by intentional doping Ga^3+^ or solubility of Ga_2_O_3_ into other metal oxides. In general, host metal oxides are In_2_O_3_ [141,156,163,164,165,166,167,222], SnO_2_ [223,224,225], and ZnO [145,226,227,228,229] for resistor gas sensors. Gas sensitive properties of Ga-doped In_2_O_3_ thin films and nanostructures were studied by Ratko et al. [222], Chen et al. [156], and Demin et al. [163,164,165,166,167]. Ga_2_O_3_ caused the formation of a porous or nanostructure in the In_2_O_3_-based ceramics, providing an active surface for reducing gases such as CH_4_ [222] and CH_2_O [156]. The response toward 100 ppm CH_2_O of Ga*_x_*In_2-*x*_O_3_ nanofiber was about four times higher than that of pure In_2_O_3_. Meanwhile, it has superior ability to selectively detect CH_2_O against other interfering VOCs. Demin et al. [163,164] obtained a high selectivity to NH_3_ against C_2_H_6_O, C_3_H_6_O and liquefied petroleum gas based on 50% In_2_O_3_-50% Ga_2_O_3_ thin film-sensitive layer. The sensitivities towards O_2_, CO, C_2_H_6_O, and CH_2_O of Ga-doped SnO_2_ materials were investigated by Silver et al. [223], Bagheri et al. [224], and Du et al. [225]. The Ga-doped SnO_2_ thin film O_2_ sensor showed sensitivity up to 2.1 for a partial pressure of oxygen as low as 1 Torr [223]. Bagheri et al. [224] observed the highest responses of 315 and 119 for 300 ppm CO and C_2_H_6_O by the SnO_2_ sensors containing 5 and 1 wt% Ga_2_O_3_, respectively. With adding more than 25 wt% Ga_2_O_3_, the sensors became selective to CO and showed negligible responses to C_2_H_6_O and CH_4_. The sensitivity to 50 ppm CH_2_O of Ga-doped SnO_2_ sensor was 4.5 times greater than that of the pure SnO_2_ sensor. Moreover, it exhibited a lower detection limit of 0.1 ppm CH_2_O with sensitivity of 3 and a short response-recovery time (3/39 s) and good selectivity. Ga-doped ZnO nanocrystalline film sensors were explored for detecting C_2_H_6_O, H_2_S, NO_2_, and H_2_. The results demonstrated that the gas response of Ga-doped ZnO sensors were greatly enhanced by compared to pristine ZnO sensor. For instance, Hou et al. [145] observed a 60% enhancement of sensing response to H_2_ by optimizing Ga composition to 0.3 at% compared with undoped ZnO sensors when measured at 130 °C. Moreover, the 0.3 at% Ga-doped ZnO sensor had a shorter response time and a better selectivity to H_2_ in a mixture of H_2_, CH_4_, and NH_3_. Rashid et al. [228] developed a 3%-Ga modified ZnO H_2_ sensor whose resistive response was improved six-fold compared with the pristine one at room temperature. It had a very low detection limit of 0.2 ppm. The enhanced H_2_S-sensing properties of Ga-doped ZnO sensors were measured by Vorobyeva et al. [227] and by Girija et al. [229]. Figure 16 shows the H_2_S response of Ga-doped ZnO gas sensors as functions of the gas concentration and temperature. In the presence of H_2_S, chemisorbed oxygen interacts with the target gas as governed by the following reaction: H_2_S + 3O^−^ → SO_2_ (g) + H_2_O (g) + 3e. The enhanced sensitivity could be attributed to both excess of oxygen vacancies due to Ga^3+^ substitution and large adsorption energy of Ga for H_2_S. Based on the liquid metal-based probe sonication route, Shafiei et al. [175] developed a quaternary GaInSnO*_x_* sensor and detection limits as low as 1 ppm and 20 ppm for NO_2_ and NH_3_ were obtained when operated at 100 °C. Table 4 summarizes the enhanced sensing performance of gas sensors using Ga-contained metal oxides toward different target gases for comparison.

#### 5.4.6. Coating with Gas Filter

If working at high temperature, Ga_2_O_3_-based gas sensors (particularly resistor-type ones), similar to other metal oxide semiconductor sensors, suffer from an issue of poor selective gas detection, since they almost react to all reducing or oxidizing gases. One of the most efficient ways to improve the selectivity of gas sensors is the use of filters [182,230]. These filters, including physical and chemical filters, are highly permeable to the target gases and can hinder interfering gases from reaching the sensors surface. Figure 17 depicts the conceptual diagrams for physical and chemical filters of Ga_2_O_3_-based gas sensors. Fleischer et al. [95] designed a selective H_2_ sensor by covering a SiO_2_ physical gas-filtering layer on the Ga_2_O_3_ sputtered film for the first time. It was found that sensors with this surface layer structure had an extremely high specificity for H_2_ upon exposure to a variety of interfering gases such as CO, CO_2_, CH_4_, C_4_H_8_, C_2_H_6_O, C_3_H_6_O, NO, and NH_3_ at 700 °C. In another work of selective H_2_, CH_4_ and NO*_x_* sensors operating at high temperature, Fleischer et al. [78] summarized that the SiO_2_ physical filter can only permeate hydrogen and the porous Ga_2_O_3_ catalytic filter removes disturbing solvent vapors by oxidation, and the gas conversion filter composed of Pt supported on Al_2_O_3_ ensures a defined NO/NO_2_ equilibrium. Flingelli et al. [77] fabricated a thin-film Ga_2_O_3_–gas sensor equipped with a screen printed porous Ga_2_O_3_ layer as catalytic filter for the selective detection of CH_4_ even in the presence of C_2_H_6_O. It was observed that the cross-sensitivities eliminated for the organic solvents were oxidized while passing the filter, and only the quite stable methane was allowed to reach the sensor surface. Weh et al. [193,194] studied the optimization of physical filters for selective high-temperature H_2_ sensors. A single SiO_2_ filter, Ga_2_O_3_/SiO_2_ and Al_2_O_3_/SiO_2_ dual filter systems, and buried filter systems in which Ga_2_O_3_ or Cr-doped SrTiO_3_ was buried between two SiO_2_ layers were applied to improve the selectivity. The results indicated that optimizing filter systems could not only increase the selectivity but also the sensitivity of a given sensor. Furthermore, these filters can be additionally used to ensure the stability over the needed lifetime of the sensor.

#### 5.4.7. Light Illumination

Light illumination rather than thermal activation is a promising strategy to enhance the gas-sensing of Ga_2_O_3_-based sensors. For clarity, the schematics and energy-level representations of the β-Ga_2_O_3_ nanowires before and after the 254 nm UV illumination are shown in Figure 18. When the sensors are illuminated under UV light with the photon energy equal or higher than the band gap of Ga_2_O_3_, electrons from the valence band can be rapidly excited to the conduction band, causing the desorption of oxygen from Ga_2_O_3_ surface and inducing the photosensitizing effect. Feng et al. [107] first reported a very fast room temperature oxygen response of the individual β-Ga_2_O_3_ nanowires synthesized by CVD under 254 nm UV illumination. This UV light-activated fast room temperature oxygen-sensing characteristic was demonstrated in the thermal evaporated β-Ga_2_O_3_ nanobelts by Ma and Fan [112]. Juan et al. [130] studied the effect on the humidity-sensing properties of a β-Ga2O3 nanowire sensor from UV light. However, they found that the humidity sensitivity with UV illumination was lower than that in the dark, since the water molecules captured the electrons and holes generated by UV light in an environment with high relative humidity. Lin et al. [155] compared the effect of UV radiation on the CO sensors made by pure Ga_2_O_3_ nanorod arrays, Pt-decorated Ga_2_O_3_ nanorod arrays, and LSFO/Ga_2_O_3_ nanorod arrays. The measured results are shown in Figure 19a,b. Under 254 nm UV illumination, the sensitivity to 100 ppm CO was enhanced by about 30%, 20%, and 50% for pristine β-Ga_2_O_3_ nanorod arrays sensors, LSFO/Ga_2_O_3_ nanorod arrays sensors and Pt decorated Ga_2_O_3_ nanorod array sensors at 500 °C, respectively. Additionally, the response times were reduced for all cases, and up to 30% reduction of response time was achieved for LSFO/Ga_2_O_3_ nanorod arrays sensors. An et al. [128] showed a significant enhancement in the response of the Pt-functionalized Ga_2_O_3_ nanorods to NO_2_ gas by UV irradiation at room temperature. It can be clearly seen from Figure 19c that the response to 5 ppm NO_2_ increases from 175% to 931% with increasing the UV light illumination intensity from 0.35 to 1.2 mW/cm^2^. A combination of the spillover effect and the enhancement of chemisorption and dissociation of gas results in the enhanced electrical response of the Pt-functionalized Ga_2_O_3_ nanostructured sensors. Sui et al. [64] realized the room temperature ozone sensing capability of InGaZnO (IGZO)-decorated amorphous Ga_2_O_3_ films under UV illumination.

## 6. Conclusions and Outlook

In conclusion, the past thirty years have witnessed very impressive progress in the field of Ga_2_O_3_-based gas sensors. It has been proven that β-Ga_2_O_3_ polycrystalline thin films and nanostructures have great potential in detecting oxygen and many reducing gases in a wide temperature range from room temperature to 1000 °C. Some growth techniques with low cost, simple process, and flexible operation are available to prepare Ga_2_O_3_-sensing material. Compared with the gas sensors made by other metal oxides such as SnO_2_, Ga_2_O_3_-based gas sensors have the advantages of long-term stability, fast response and recovery times, good reproducibility, low cross-sensitivity to water vapor, and short preaging time [181]. Figure 20 displays the radar chart of seven performance enhancement strategies of gas sensors made by Ga_2_O_3_ bulk crystal, thin films, and nanostructures. Although the performance of Ga_2_O_3_-based gas sensors has been improved through various enhancement strategies, from a practical point of view, continuous efforts should be made to further increase the gas selectivity, to decrease the operating temperature, and to develop high-speed nonresistive types of sensors. In-depth understanding of the interaction between the target gas and Ga_2_O_3_ surface, as well as the gas-sensing behavior, are also needed using newly developed computational tools. Some suggestions on future research opportunities of Ga_2_O_3_-based gas sensors are listed below:
(i)Construction of hybrid structures with two-dimensional (2D) materials and organic polymers. Two-dimensional materials are regarded as having promising potential for gas-sensing devices owing to their large surface-to-volume ratio and high surface sensitivity [231]. The integration of 2D layers with Ga_2_O_3_, especially a 2D Ga_2_O_3_ monolayer [232], can form a van der Waals heterojunction without constraints on the chemical bonding and interfacial lattice matching, which will be expected to widen the building blocks for novel applications with unprecedented properties and excellent performance. The energy band alignment of such van der Waals heterojunction could be precisely designed by selecting suitable 2D materials, such as transition metal dichalcogenides with different band gaps and working functions so as to optimize the gas selectivity and performance of the Ga_2_O_3_ gas sensor. Additionally, organic semiconductors offer a viable alternative to conventional inorganic semiconductors in gas sensor applications because of their unusual electrical properties, diversity, large area, and potentially low cost [233]. They show good sensitivities toward many gases or vapors, ranging from organic solvents to inorganic gases. Moreover, it allows for the inexpensive fabrication of novel Ga_2_O_3_ gas sensors using organic polymers as a platform, owing to their porosity, mechanical flexibility, environmental stability, and solution processability. Therefore, more research should be conducted to fabricate Ga_2_O_3_-based gas sensors by constructing heterostructures with 2D materials and organic polymers, which may result in some novel features and potential applications.(ii)Combinations with DFT calculations and machine learning. Hitherto, quite few theoretical investigations have been devoted to understanding the interaction behaviors between gas molecules and Ga_2_O_3_ in the field of gas sensors. DFT calculations [232,234,235] have been performed to study the adsorption, dissociation, and diffusion characteristics of gas molecules on the surface of Ga_2_O_3_. It is suggested that the calculated adsorption energy, electron structures of oxygen vacancy, and charge transfer are highly significant in defining the performance of a sensor. Large adsorption energy and significant charge transfer indicate high sensitivity and selectivity towards a specific gas. Alongside DFT, machine learning is considered as an effective data processing approach for developing smart devices with the ability to deal with selectivity and drift problems [236]. The machine learning technique involves data processing of sensor output, dimensionality reduction, and then training a system/network for the predictions. Therefore, DFT and machine learning can be implemented as a powerful tool for studying gas-sensing behavior and provide valuable suggestions and predictions of gas-sensing materials and target gas species.(iii)Development of optical sensors. Compared to electrical characteristic gas sensors, more attention should be paid on the less-explored sensors using the promising optical properties of Ga_2_O_3_. In a pioneer study on Ga_2_O_3_-based optical gas sensors, Reiprich et al. [122] pointed out that the spectral composition of photoluminescence showed a strong intensity difference upon exposure to various gases at room temperature indirectly caused by the negatively charged oxygen ions in Ga_2_O_3_. Optical gas sensors offer a number of advantages, such as fast responses, minimal drift and high gas specificity [187]. Gas detection can be made in real time and in situ by adopting optical sensors. With the help of the rapid developed technology of Ga_2_O_3_ optical devices, it is possible to design Ga_2_O_3_-based optical gas sensors that can measure gas concentrations in the ppm or ppb range with zero cross-response to other gases and high temporal resolution. In this way, optical gas-sensing fills a gap between low-cost electrical/electrochemical sensors with inferior performance and high-end analytic equipment.

Moreover, the researchers should explore the options for the use of Ga_2_O_3_ in many other sensor applications such as chemical sensors [237,238] and biosensors [239,240] to detect metal ions, e.g., Ni^2+^ and Fe^2+^, and DNA sequences in biological fluids, tap water, and so forth. It is believed that Ga_2_O_3_ will play a significant role in the field of safety, environmental, and medical monitoring systems with the development of device designs and fabrication technologies.

## Figures and Tables

**Figure 1 materials-15-07339-f001:**
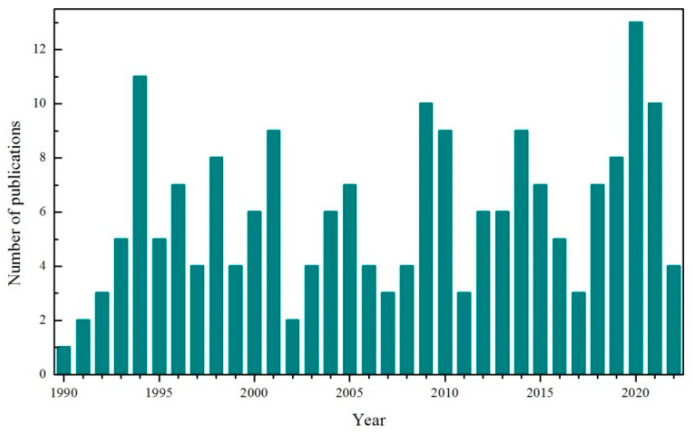
Number of publications on Ga_2_O_3_-based gas sensors from 1990 to 2022. Data extracted from Web of Science with keywords “sensor” or “sensing” and “Ga_2_O_3_” or “gallium oxide”.

**Figure 2 materials-15-07339-f002:**
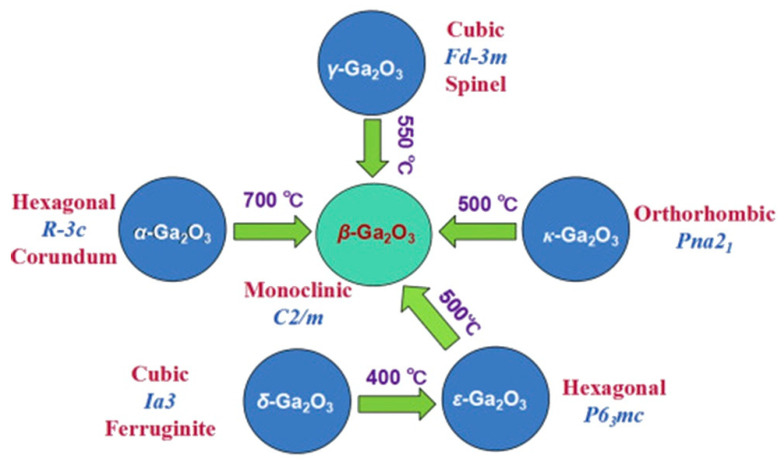
Interconversion relation of Ga_2_O_3_ polymorphs [22]. Copyright 2019 Elsevier.

**Figure 3 materials-15-07339-f003:**
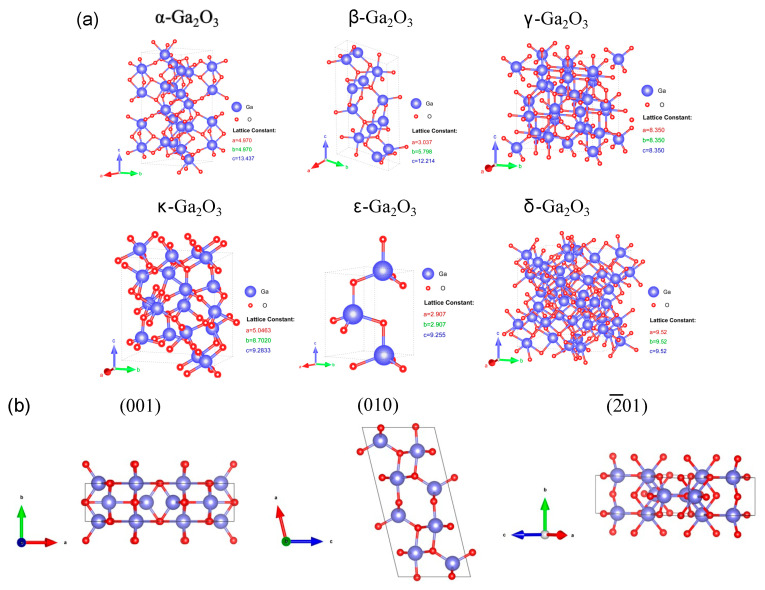
(**a**) Crystal structures of six polymorphs of Ga_2_O_3_. (**b**) The (001), (010), and ($\bar{2}$01) planes for β-Ga_2_O_3_. Original drawing using VESTA was done by us.

**Figure 4 materials-15-07339-f004:**
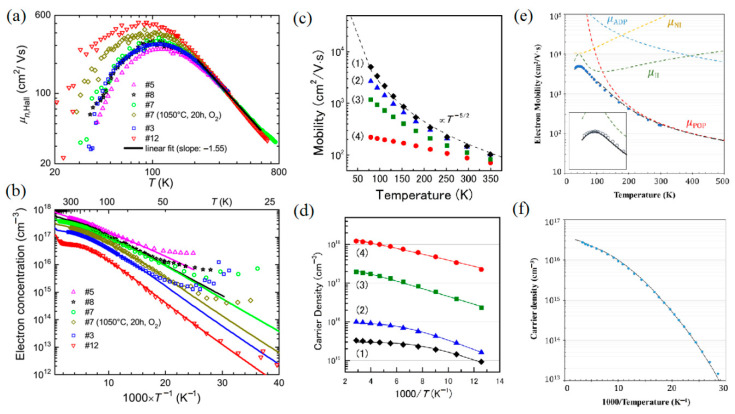
(**a**) The Hall mobility and (**b**) electron density as functions as the temperature for β-Ga_2_O_3_ single crystals grown by the Czochralski method [58]. Copyright 2011 American Institute of Physics. (**c**) The Hall mobility and (**d**) electron density as functions as the temperature for Si-doped homoepitaxial films grown on β-Ga_2_O_3_ (001) substrate by halide vapor-phase epitaxy [60]. Copyright 2018 Elsevier. (**e**) The Hall mobility and (**f**) electron density as functions as the temperature for silicon-doped β-Ga_2_O_3_ homoepitaxial films grown on β-Ga_2_O_3_ (010)-oriented substrates via metalorganic chemical vapor deposition [56]. Copyright 2019 American Institute of Physics.

**Figure 5 materials-15-07339-f005:**
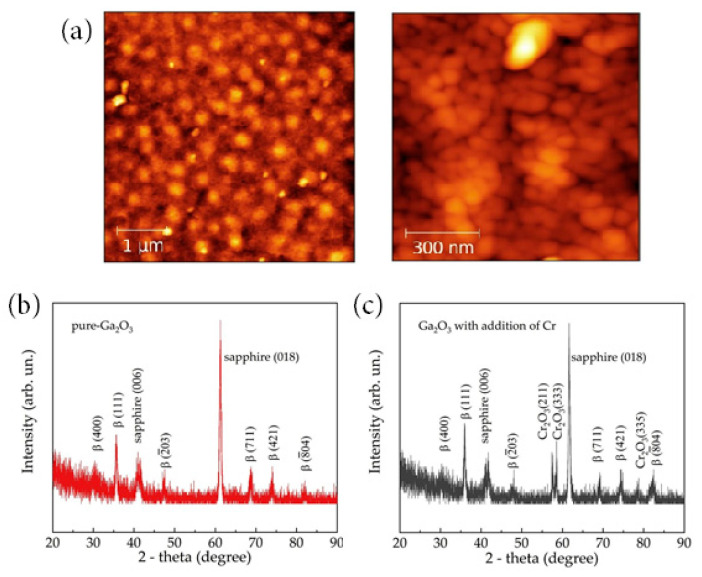
(**a**) AFM images of the Cr-doped Ga_2_O_3_ sputtering thin film [87]. XRD patterns of (**b**) pure Ga_2_O_3_ thin films and (**c**) Cr-doped Ga_2_O_3_ thin films [87]. Copyright 2020 Elsevier.

**Figure 7 materials-15-07339-f007:**
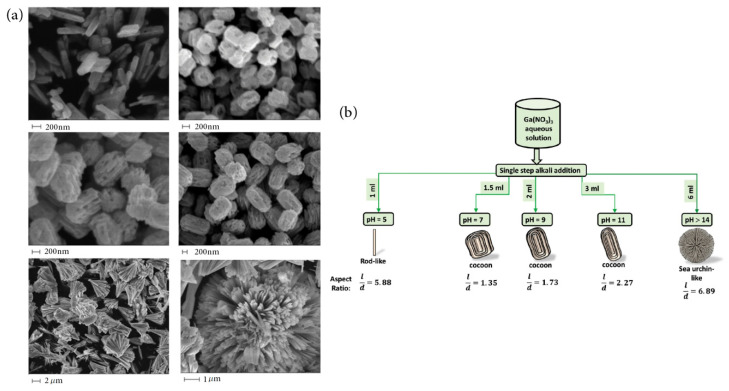
(**a**) SEM images of the morphological evolution of β-Ga_2_O_3_ nanostructures [159]. (**b**) Schematic representation of morphological evolutions at different pH values [159]. Copyright 2020 Elsevier.

**Figure 8 materials-15-07339-f008:**
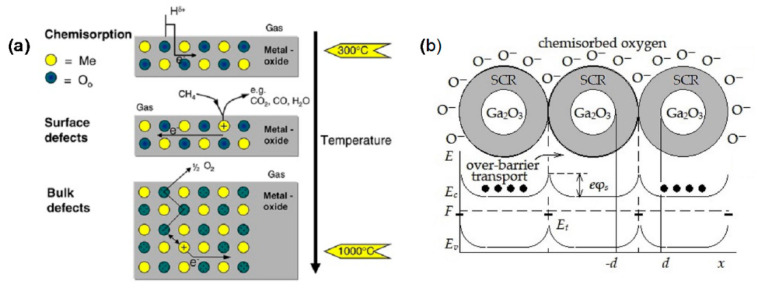
(**a**) The model of the three temperature-dependent regimes of the gas reaction of Ga_2_O_3_ [182]. Copyright 2008 IOP. (**b**) The model and corresponding energy diagram of Ga_2_O_3_ grain contact [87]. Copyright 2020 Elsevier.

**Figure 9 materials-15-07339-f009:**
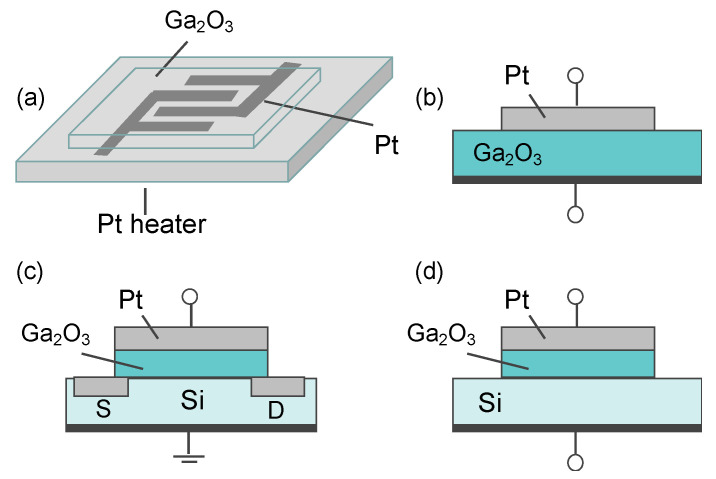
Schematic device structures for (**a**) resistor-type (**b**) SBD-type, (**c**) FET-type, and (**d**) capacitor-type Ga_2_O_3_-based gas sensors. Redrawn with permission from G. Korotcenkov [187]. Copyright 2014 Springer.

**Figure 10 materials-15-07339-f010:**
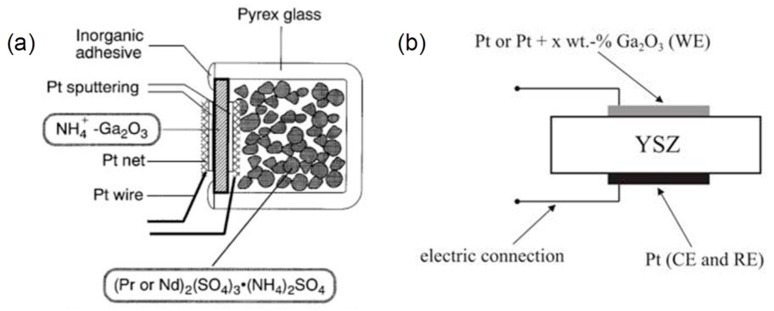
Schematic device structures for electrochemical gas sensors: (**a**) NH^4+^-Ga_2_O_3_ solid electrolyte [210] Copyright 1998 The Electrochemical Society. (**b**) YSZ solid electrolyte [214]. Copyright 2004 Elsevier.

**Figure 11 materials-15-07339-f011:**
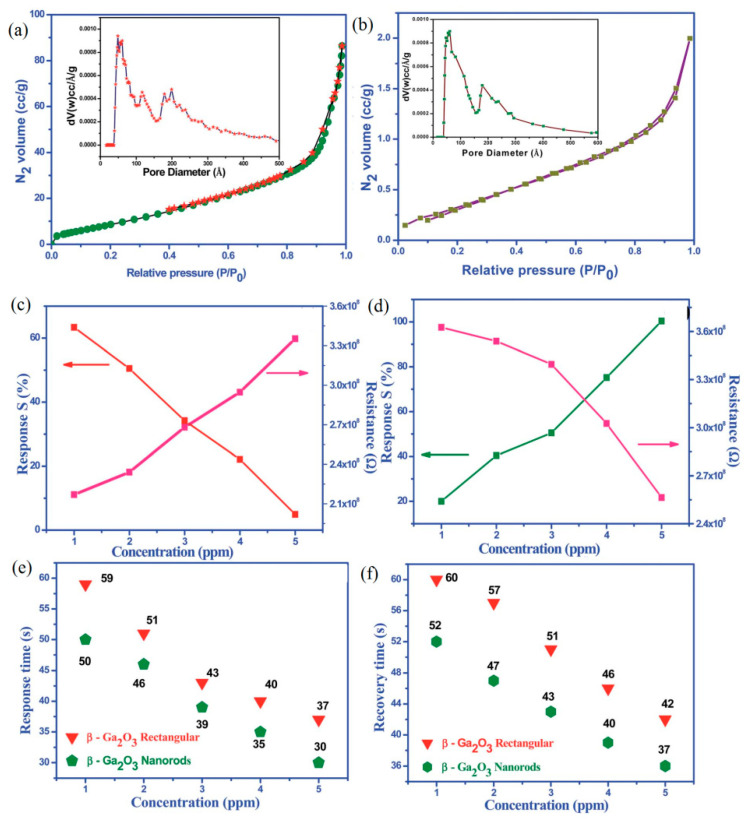
Nitrogen adsorption-desorption isotherm of the Ga_2_O_3_ (**a**) rectangular and (**b**) rod-shaped nanostructures (inset: the corresponding pore size distribution). Sensing response and electrical resistance as a function of CO concentration at 100 °C (**c**) β-Ga_2_O_3_ rectangular nanorods, (**d**) β-Ga_2_O_3_ nanorods, (**e**) response time, and (**f**) recovery time of different β-Ga_2_O_3_ nanostructure thin films [135]. Copyright 2016 The Royal Society of Chemistry.

**Figure 12 materials-15-07339-f012:**
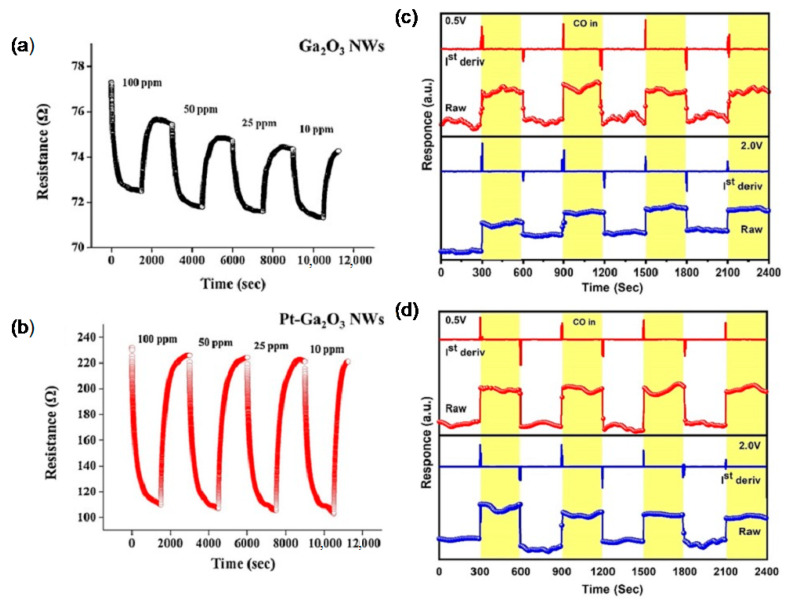
Gas responses of (**a**) bare Ga_2_O_3_ nanowires and (**b**) Pt-coated Ga_2_O_3_ nanowires to 10, 25, 50, and 100 ppm CO gas at 100 °C [124]. Copyright 2012 Elsevier. Room temperature CO gas sensor measurement results with different bias voltages for (**c**) pure β-Ga_2_O_3_ nanowires and (**d**) Au-decorated β-Ga_2_O_3_ nanowire with 2158 ppm CO concentration [140]. Copyright 2020 Elsevier.

**Figure 13 materials-15-07339-f013:**
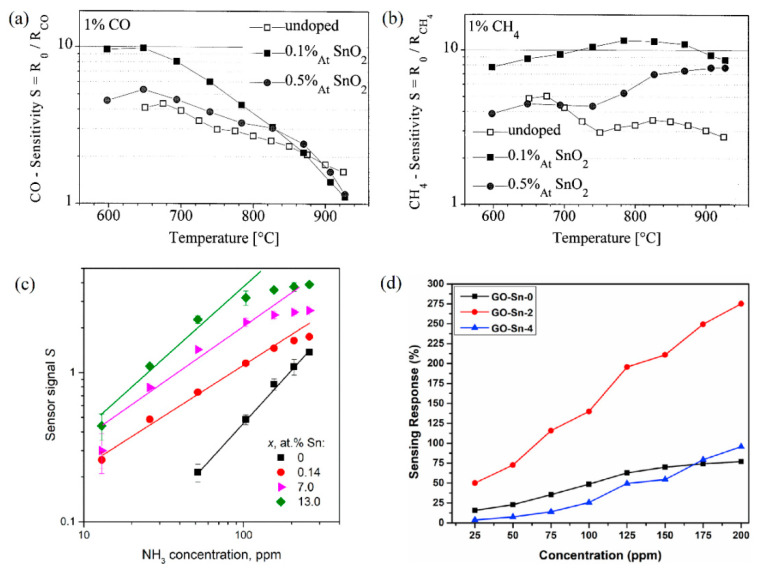
Enhancement of the sensitivity to (**a**) 1% CO and (**b**) 1% CH_4_ of Sn-doped Ga_2_O_3_ thin films compared to undoped films [189]. Copyright 1998 Elsevier. (**c**) Sensor signal as a function of NH_3_ concentration for Sn-doped Ga_2_O_3_ samples measured at 500 °C [170]. Copyright 2021 MDPI. (**d**) Room temperature-sensing response as a function of NH_3_ concentration [162]. Copyright 2021 Elsevier.

**Figure 14 materials-15-07339-f014:**
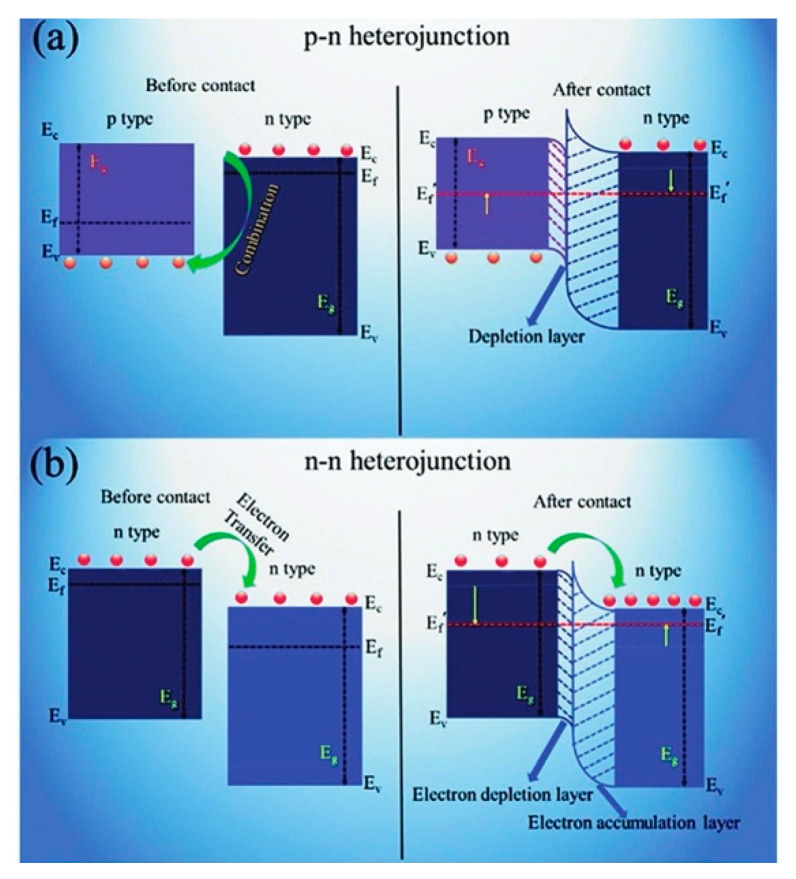
Schematic illustrations of the energy band structures at heterojunction interfaces of (**a**) p–n and (**b**) n–n of heterojunctions [178]. 2019 The Royal Society of Chemistry.

**Figure 15 materials-15-07339-f015:**
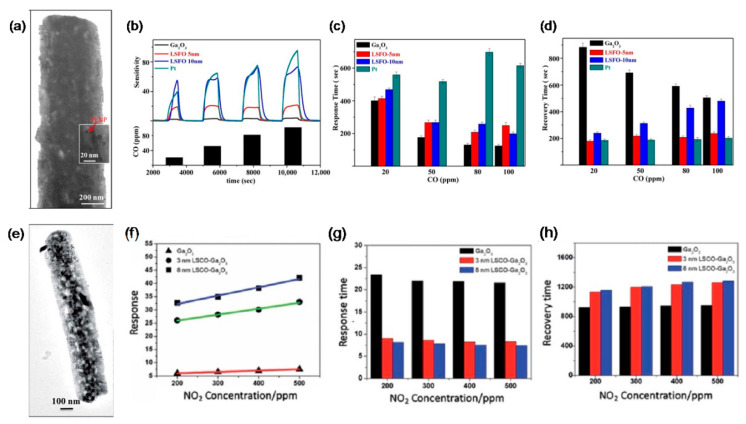
(**a**) TEM image of a β-Ga_2_O_3_ nanorod coated with 5 nm LSFO. (**b**) Sensitivity, (**c**) response time, and (**d**) recover time of gas sensors made by β-Ga_2_O_3_/LSFO nanorods upon exposure to CO at 500 °C [154]. Copyright 2016 American Chemical Society. (**e**) TEM image of a β-Ga_2_O_3_ nanorod coated with 8 nm LSCO. (**f**) Sensitivity, (**g**) response time, and (**h**) recover time of gas sensors made by β-Ga_2_O_3_/LSCO nanorods upon exposure to NO_2_ at 800 °C [161]. Copyright 2020 The Royal Society of Chemistry.

**Figure 16 materials-15-07339-f016:**
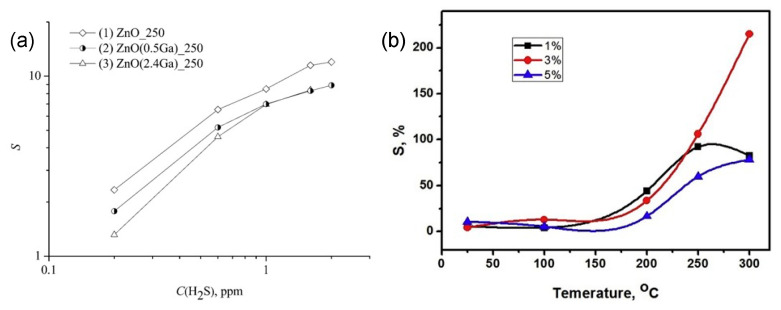
(**a**) H_2_S Response of Ga-doped ZnO gas sensors as a function of the gas concentration [227]. Copyright 2013 Elsevier. (**b**) H_2_S Response of Ga-doped ZnO gas sensors as a function of the temperature [229]. Copyright 2018 Elsevier.

**Figure 17 materials-15-07339-f017:**
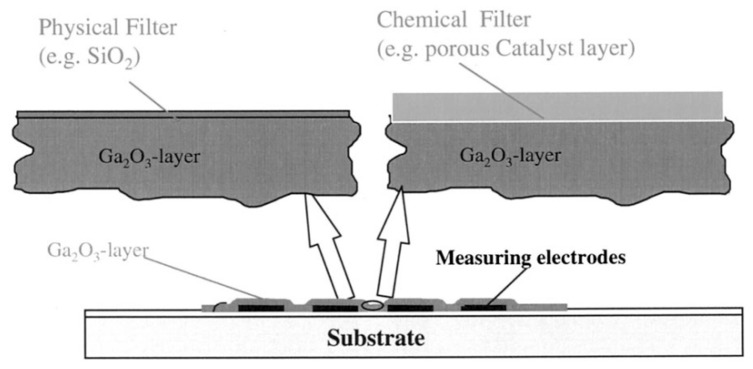
Conceptual diagrams for physical and chemical filters of Ga_2_O_3_-based gas sensors [230]. Copyright 1998 Elsevier.

**Figure 18 materials-15-07339-f018:**
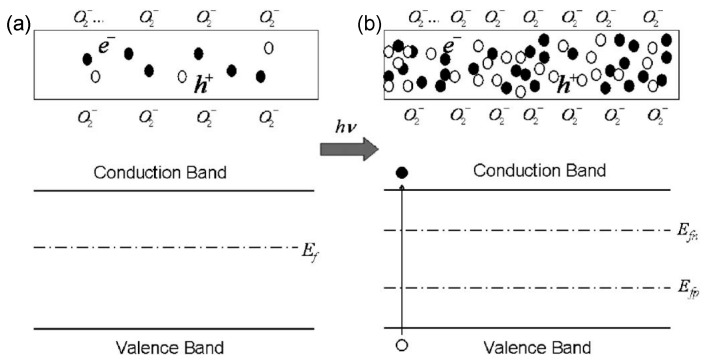
Schematics and energy-level representations of the β-Ga_2_O_3_ nanowires (**a**) before and (**b**) after the 254 nm UV illumination [107]. Copyright 2006 American Institute of Physics.

**Figure 19 materials-15-07339-f019:**
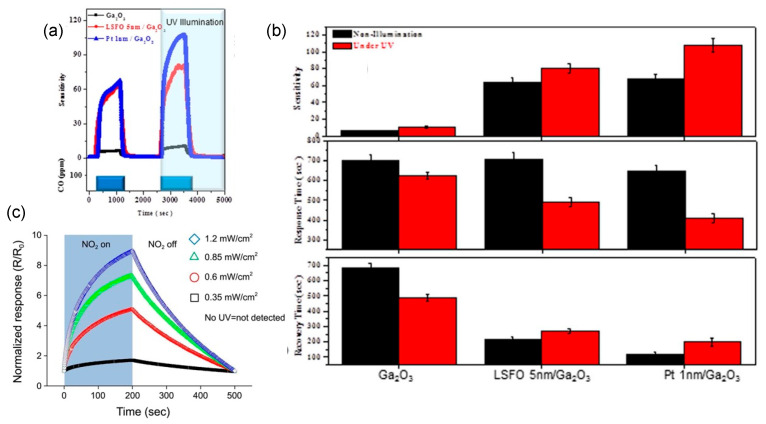
CO gas-sensing performance comparison of pristine, LSFO-decorated, and Pt-decorated β-Ga_2_O_3_ nanorod arrays: (**a**) normalized sensitivity time characteristics and (**b**) sensitivity, recovery time, and recovery time under dark or UV illumination tested at 500 °C [155]. Copyright 2017 American Institute of Physics. (**c**) Room temperature gas responses of Pt-functionalized Ga_2_O_3_ nanorod gas sensors to 5 ppm NO_2_ under UV light illumination with varied intensities [128]. Copyright 2013 Korean Chemical Society.

**Figure 20 materials-15-07339-f020:**
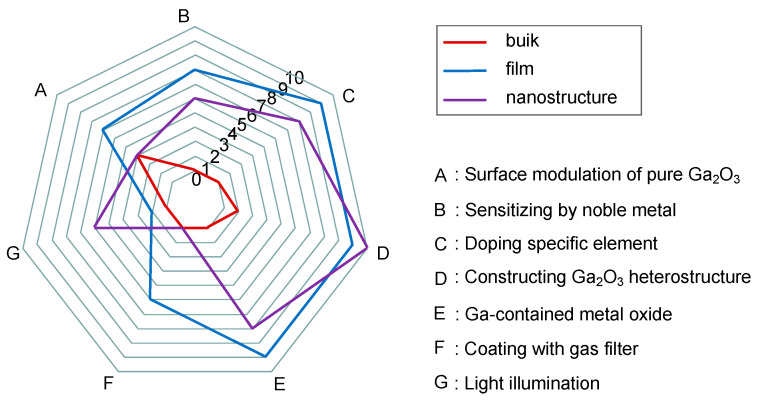
Radar chart of seven performance enhancement strategies of gas sensors made by Ga_2_O_3_ bulk crystal, thin films, and nanostructures. Original drawing was done by us.

**Table 1 materials-15-07339-t001:** Basic properties of Ga_2_O_3_ polymorphs [22,32].

Polymorph	System	Space Group	Lattice Parameters
α	hexagonal	$R\bar{3}c$	*a* = *b* = 4.98–5.04 Å, *c* = 13.4–13.6 Å, *α* = *β* = 90°, *γ* = 120°
β	monoclinic	C2/m	*a* = 12.12–12.34 Å, *b* = 3.03–3.04 Å, *c* = 5.80–5.87 Å, *α* = *β* = 90°, *γ* = 103.8°
γ	cubic	$\mathrm{Fd}\bar{3}m$	*a* = *b* = *c* = 8.24–8.30 Å, *α* = *β* = *γ* = 90°
δ	cubic	Ia3	*a* = *b* = *c* = 9.40–10.1 Å, *α* = *β* = *γ* = 90°
ε	hexagonal	P63mc	*a* = 5.06–5.12 Å, *b* = 8.69–8.79 Å, *c* = 9.3–9.4 Å, *α* = *β* = 90°, *γ* = 120°
κ	orthorhombic	Pna2_1_	*a* = 5.05 Å, *b* = 8.69 Å, *c* = 9.27 Å, *α* = *β* = *γ* = 90°

**Table 2 materials-15-07339-t002:** Basic electric properties of β-Ga_2_O_3_ [32,61].

Electric Properties	Value
Electronic effective mass (*m*_0_)	0.28
Static dielectric constant	10
High frequency dielectric constant	3.9
Electron mobility (cm^2^·V^−1^·s^−1^)	200
Range of free electron concentration (cm^−3^)	10^16^~10^20^
Range of doping concentration (cm^−3^)	10^17^~10^20^
Electron affinity (eV)	4.0
Break down field (eV/cm)	8.0
Typical types of shallow donors	Sn, Ge, Si

**Table 3 materials-15-07339-t003:** Target gases of Ga_2_O_3_-based gas sensors.

	Electrical Gas Sensors				Electrochemical Gas Sensor	Optical Gas Sensor
	Resistor	SBD	FET	Capacitor		
Environmental gases	O_2_, CO_2_, O_3_, NH_3_, SO_2_	O_2_, NH_3_	NH_3_		NH_3_	
Highly toxic gases	CO, H_2_S, NO, NO_2_	CO, NO	CO, NO_2_		CO	
Combustible gases	H_2_, CH_4_, C_4_H_10_, C_3_H_8_, C_4_H_8_, C_7_H_8_	H_2_, CH_4_, C_3_H_6_	H_2_	C_7_H_8_	C_3_H_6_	
VOCs	C_2_H_6_O, C_3_H_6_O, C_3_H_8_O			C_2_H_6_O, C_3_H_6_O, C_3_H_8_O		C_2_H_6_O, C_3_H_6_O, C_3_H_8_O
Humidity	H_2_O					
Other gases	C_2_H_6_S			CH_3_NO_2_, C_6_H_15_N		

**Table 4 materials-15-07339-t004:** Enhanced sensing performances of gas sensors using Ga-contained metal oxides.

Sensing Material	Preparation	Sensitivity @Gas Concentration	Operating Temperature	Response/Recover Time (s)	Other Observations	Reference
Ga-doped In_2_O_3_ nanowire	CVD	^△^1.05 @ 80 ppm C_2_H_6_O	200	40/800		[141]
		^△^2.2 @ 4 ppm NO_2_	200	1980/2780		
Ga-doped In_2_O_3_ nanofiber	Hydrothermal synthesis	^◊^52.5 @ 100 ppm CH_2_O	150	1/70	The low limit of detection is 0.2 ppm	[156]
Ga-doped In_2_O_3_ film	PLD	^□^2.15 @ 25 ppm NH_3_	623			[164]
		^□^20.5 @ 25 ppm C_2_H_6_O	504			
		^□^24.4 @ 25 ppm C_3_H_6_O	504			
		^□^7.47 @ 25 ppm CH_4_	500			[167]
Ga-doped In_2_O_3_ ceramics	Coprecipitation	^△^0.85 @ 0.5% CH_4_	380			[222]
Ga-doped SnO_2_ film	Spray pyrolysis	^◊^3.1 @ 1 Torr O_2_	350			[223]
Ga-doped SnO_2_ nanocomposites	Coprecipitation	^◊^315 @ 300 ppm CO	300	36		[224]
Ga-doped SnO_2_ nanocomposites		^◊^119 @ 300 ppm C_2_H_6_O	250	93		
Ga-doped SnO_2_ microflowers	Hydrothermal synthesis	^◊^95.8 @ 50 ppm CH_2_O	230	3	The low limit of detection is 0.1 ppm	[225]
Ga-doped ZnO film	Sol–gel synthesis	^△^0.5 @ 500 ppm H_2_	130	475		[145]
Ga-doped ZnO Nanorod	Hydrothermal synthesis	^△^1.01% @ 250 ppm C_2_H_6_O	RT			[226]
Ga-doped ZnO nanoparticle	Spray pyrolysis	^△^56 @ 2 ppm NO_2_	250			[227]
		^△^8 @ 1 ppm H_2_S	250			
Ga-doped ZnO nanorod	Sol–gel synthesis	^△^0.91 @ 100 ppm H_2_	RT	20		[228]
Ga-doped ZnO film	Magnetron sputtering	^△^2.4 @ 5 ppm H_2_S	300			[229]

^△^ $S=\left(Ra-Rg\right)/\mathrm{Ra}$; ^◊^S=Ra/Rg; ^□^$S=\left(Ra-Rg\right)/Rg$

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
