# Peer review of "Gallium Oxide for Gas Sensor Applications: A Comprehensive Review"

_materials, 2022, doi:10.3390/ma15207339_

Round 1

Reviewer 1 Report

The authors have developed in depth every aspect concerning gallium oxide for gas sensors and in presenting the properties and production methods, they have always taken into account their impact on their use in the field of sensors. In a first analysis may appear too heavy for the reader because it is full of information but being a review, this aspect is fine too.

In my opinion this manuscript can be published in Materials.

-What is the main question addressed by the research?

The authors addressed every aspect concerning gallium oxide for gas sensors, from the crystallography of the material to the production methods, from the production technologies to the strategies for obtaining an improvement in the performance of the devices based on this material.

-Is it relevant and interesting?

In my opinion, it is interesting for those who intend to use this material because it becomes a means of orientation also with respect to scientific literature.

-How original is the topic?

Certainly, there are other similar reviews that deal with the topic of gas sensors based on gallium nitride, but having an updated review is always useful for a researcher working in this field.

-What does it add to the subject area compared with other published material?

The interesting aspect is that it is a real compendium, as it deals with all the possible arguments attributable to gas sensors based on gallium nitride

-Is the paper well written? Is the text clear and easy to read?

 The work is well written and structured, which is not easy given the complexity of the addressed topics. Of course, the simplicity of reading is for who deal with gas sensors.

-Are the conclusions consistent with the evidence and arguments presented? Do they address the main question posed?

Although the authors had tried to give an opinion even also on the others’ work, they also add more streamlined conclusions just to help those who got lost in the enormous amount of presented information.

So, in my opinion, they have succeeded in their intent, that is to put together works on this issue to show the robustness of this material to be used in the field of gas sensors.

Author Response

The authors thank for the reviewer's instructive comments. There is no change in the revised manuscript according to the review report.

Reviewer 2 Report

The current manuscript is giving overview about the synthesis, characterization and utlization of Gallium oxide for gas sensor applications. The review is well written and organized and giving comprehensive information.

I would recommend for publication. However, authors have to enrich the introduction section with section about different sensors application and possibility of using Gallium oxide as electroactive materials for different sensor for detection of heavy metals and other pharmaecutical molecules.

Then,

I would citation of the following reports

Analytica Chimica Acta 1197 (2022) 339518

Microporous and Mesoporous Materials 313 (2021)110832

Author Response

The authors thank for the reviewer's instructive comments. In the revised manuscript, we added a paragraph to mention the other sensor application of Ga2O3 in the end. In this paragraph, we added the citation of the articles suggested by the reviewer. 

Reviewer 3 Report

1.Authors should refer also the article recently published

"Recent progress of Ga2O3-based gas sensors" Hongchao Zhai, Zhengyuan Wu *, Zhilai Fang ,https://doi.org/10.1016/j.ceramint.2022.06.066

2. page.5 in the table 2 modify for "breack down field"

Author Response

The authors thank for the reviewer's instructive comments. In the revised manuscript, we replaced Ref. 27 with the newly article suggested by the reviewer. This was also mentioned in the last sentence in on Page 1. The "break field" in Table 2 was corrected as "break down field".